# TREE SEARCH-BASED POLICY OPTIMIZATION UNDER STOCHASTIC EXECUTION DELAY

**David Valensi**
Technion
davidvalensi@campus.technion.ac.il

**Esther Derman**
Technion
estherderman@campus.technion.ac.il

**Shie Mannor**
Technion & Nvidia Research
shie@ee.technion.ac.il

**Gal Dalal**
Nvidia Research
gdalal@nvidia.com

## ABSTRACT

The standard formulation of Markov decision processes (MDPs) assumes that the agent's decisions are executed immediately. However, in numerous realistic applications such as robotics or healthcare, actions are performed with a delay whose value can even be stochastic. In this work, we introduce stochastic delayed execution MDPs, a new formalism addressing random delays without resorting to state augmentation. We show that given observed delay values, it is sufficient to perform a policy search in the class of Markov policies in order to reach optimal performance, thus extending the deterministic fixed delay case. Armed with this insight, we devise DEZ, a model-based algorithm that optimizes over the class of Markov policies. DEZ leverages Monte-Carlo tree search similar to its non-delayed variant EfficientZero to accurately infer future states from the action queue. Thus, it handles delayed execution while preserving the sample efficiency of EfficientZero. Through a series of experiments on the Atari suite, we demonstrate that although the previous baseline outperforms the naive method in scenarios with constant delay, it underperforms in the face of stochastic delays. In contrast, our approach significantly outperforms the baselines, for both constant and stochastic delays. The code is available at https://github.com/davidva1/Delayed-EZ.

## 1 INTRODUCTION

The conventional Markov decision process (MDP) framework commonly assumes that all of the information necessary for the next decision step is available in real time: the agent's current state is immediately observed, its chosen action instantly actuated, and the corresponding reward feedback concurrently perceived (Puterman, 2014). However, these input signals are often delayed in real-world applications such as robotics (Mahmood et al., 2018), healthcare (Politi et al., 2022), or autonomous systems, where they can manifest in different ways. Perhaps the most prominent example where delay can be stochastic is in systems that rely on data transmission. Oftentimes, there is some interference in transmission that stems from internal or external sources. Internal sources may be due to temperature or pressure conditions that influence the sensing hardware, whereas external sources of interference occur when a policy infers actions remotely (e.g., from a cloud). For example, an autonomous vehicle may experience delay from its perception module in recognizing the environment around it. This initial recognition delay is known as *observation delay*. Additional delays can occur when taking action according to a previously made decision. This delay in response is termed *execution delay*. As highlighted in (Katsikopoulos & Engelbrecht, 2003) and despite their distinct manifestations, both types of delay are functionally equivalent and can be addressed using similar methodologies.

In addition to its existence, the delay's nature is often overlooked its nature. In complex systems, delays exhibit stochastic properties that add layers of complexity to the decision-making process (Dulac-Arnold et al., 2019). This not only mirrors their unpredictability, but also warrants a novel

approach to reinforcement learning (RL) that does not rely on state augmentation. This popular method consists of augmenting the last observed state with the sequence of actions that have been selected by the agent, but whose result has not yet been observed (Bertsekas, 2012; Altman & Nain, 1992; Katsikopoulos & Engelbrecht, 2003). Although it presents the advantage of recovering partial observability, this approach has two major limitations: (i) its computational complexity inevitably grows exponentially with the delay value (Derman et al., 2021); (ii) it structurally depends on that value, which prevents it from generalizing to random delays. In fact, the augmentation method introduced in (Bouteiller et al., 2020) to address random delays was practically tested on small delay values and/or small support.

In this work, we tackle the following question: How does one effectively engage in an environment where action repercussions are subject to random delays? We introduce the paradigm of stochastic execution delay MDPs (SED-MDPs). We then establish a significant finding: To address stochastic delays in RL, it is sufficient to optimize within the set of Markov policies, which is exponentially smaller than that of history-dependent policies. Our result improves upon the one in (Derman et al., 2021) that tackled the narrower scope of deterministic delays.

Based on the observation above, we devise *Delayed EfficientZero* (DEZ). This model-based algorithm builds on the strengths of its predecessor, EfficientZero, by using Monte Carlo tree search to predict future actions (Ye et al., 2021). In practice, DEZ keeps track of past actions and their delays using two separate queues. It utilizes these queues to infer future states and make decisions accordingly. We also improve the way the algorithm stores and uses data from previous experience, enhancing its overall accuracy. In essence, DEZ offers a streamlined approach to managing stochastic delays in decision-making. We accordingly modify the Atari suite to incorporate both deterministic and stochastic delays, where the delay value follows a random walk. In both cases, our algorithm surpasses 'oblivious' EfficientZero that does not explicitly account for delay, and 'Delayed-Q' from Derman et al. (2021).

In summary, our contributions are as follows.

1. We formulate the framework of MDPs with stochastic delay and address the principled approach of state augmentation.

2. We prove that if the realizations of the delay process are observed by the agent, then it suffices to restrict policy search to the set of Markov policies to attain optimal performance.

3. We devise DEZ, a model-based algorithm that builds upon the prominent EfficientZero. DEZ yields non-stationary Markov policies, as expected by our theoretical findings. It is agnostic to the delay distribution, so no assumption is made about the delay values themselves. Our approach is adaptable and can be seamlessly integrated with any of the alternative model-based algorithms.

4. We thoroughly test DEZ on the Atari suite under both deterministic and stochastic delay schemes. In both cases, our method achieves significantly higher reward than the original EfficientZero and 'Delayed-Q' from Derman et al. (2021).

## 2 RELATED WORK

Although it is critical for efficient policy implementation, the notion of delayed execution remains scarce in the RL literature. One way to address this is through state-space augmentation, which consists of concatenating all pending actions to the original state. This brute-force method presents the advantage of recovering the Markov property, but its computational cost increases exponentially with the delay value (Walsh et al., 2009; Derman et al., 2021).

Previous work that addressed random delay using state embedding includes Katsikopoulos & Engelbrecht (2003); Bouteiller et al. (2020). The work Katsikopoulos & Engelbrecht (2003) simply augments the MDP with the maximal delay value to recover all missing information. In (Bouteiller et al., 2020), such an augmented MDP is approximated by a neural network, and trajectories are resampled to compensate for the actual realizations of the delay process during training. The proposed method is tested on a maximal delay of 6 with a high likelihood near 2. This questions the viability of their approach to (i) higher delay values; and (ii) delay distributions that are more evenly distributed over the support. Differently, by assuming the delay process to be observable, our method

augments the state space by one dimension only, regardless of the delay value. It also shows efficiency for delay values of up to 25, which we believe comes from the agnosticism of our network structure to the delay.

To avoid augmentation, Walsh et al. (2009) alternatively proposes inferring the most likely present state and deterministically transitioning to it. Their model-based simulation (MBS) algorithm requires the original transition kernel to be almost deterministic for tractability. Additionally, MBS proceeds offline and requires a finite or discretized state space, which raises the curse of dimensionality. (Derman et al., 2021) address the scalability issue through their Delayed DQN algorithm, which presents two main features: (i) In the same spirit as MBS, Delayed DQN learns a forward model to estimate the state induced by the current action queue; (ii) This estimate is stored as a next-state observation in the replay buffer, thus resulting in a time shift for the Bellman update. Although (Walsh et al., 2009; Derman et al., 2021) avoid augmentation, these two works focus on a fixed and constant delay value, whereas DEZ allows it to be random. In a related study, Karamzade et al. (2024) investigate a method similar to ours, focusing on continuous control tasks in the mujoco and DMC environments. Notably, they adopt our latent space imagination approach in their adaptation of Dreamer-V3 to manage delays.

DEZ leverages the promising results obtained from tree-search-based learning and planning algorithms (Schrittwieser et al., 2020; Ye et al., 2021). Based on a Monte Carlo tree search (MCTS), MuZero (Schrittwieser et al., 2020) and EfficientZero (Ye et al., 2021) perform multi-step lookahead and in-depth exploration of pertinent states. This is performed alongside latent space representations to effectively reduce the state dimensions. MuZero employs MCTS as a policy improvement operator by simulating multiple hypothetical trajectories within the latent space of a world model. On the other hand, EfficientZero builds upon MuZero's foundation by introducing several enhancements. These enhancements include a self-supervised consistency loss, the ability to predict returns over short horizons in a single step, and the capacity to rectify off-policy trajectories using its world model. Yet, none of these methods account for the delayed execution of prescribed decisions, whereas DEZ takes advantage of the world model to infer future states and accordingly adapt decision-making.

## 3 Preliminaries

A discounted infinite-horizon MDP is a tuple $(\mathcal{S}, \mathcal{A}, P, r, \mu, \gamma)$ where $\mathcal{S}, \mathcal{A}$ are finite state and action spaces, respectively, $P$ is a transition kernel $P : \mathcal{S} \times \mathcal{A} \to \Delta_{\mathcal{S}}$ that maps each state-action pair to a distribution over the state space, $r : S \times A \to \mathbb{R}$ is a reward function, $\mu \in \Delta_{\mathcal{S}}$ an initial state distribution, and $\gamma \in [0, 1)$ a discount factor that diminishes the weight of long-term rewards.

At each decision step $t$, an agent observes a state $\tilde{s}_t$, draws an action $\tilde{a}_t$ that generates a reward $r(\tilde{s}_t, \tilde{a}_t)$, then progresses to $\tilde{s}_{t+1}$ according to $P(\cdot|\tilde{s}_t, \tilde{a}_t)$. The action $\tilde{a}_t$ follows some prescribed decision rule $\pi_t$, which itself has different possible properties. More precisely, a decision rule can be Markovian (M) or history-dependent (H) depending on whether it takes the current state or the entire history as input. It can be randomized (R) or deterministic (D) depending on whether its output distribution supports multiple or single action. This yields four possible classes of decision rules: HR, MR, HD, MD. A policy $\pi := (\pi_t)_{t \in \mathbb{N}}$ is a sequence of decision rules. The different classes of policies are denoted by $\Pi^{\mathrm{HR}}, \Pi^{\mathrm{MR}}, \Pi^{\mathrm{HD}}, \Pi^{\mathrm{MD}}$ and determined by their decision rules. When $\pi_t = \pi_0, \forall t \in \mathbb{N}$, the resulting policy is said to be stationary. We simply denote by $\Pi$ the set of stationary policies.

The value function under a given policy $\pi \in \Pi^{\mathrm{HR}}$ is defined as

$$v^\pi(s) := \mathbb{E}^\pi \left[ \sum_{t=0}^{\infty} \gamma^t r(\tilde{s}_t, \tilde{a}_t) | \tilde{s}_0 = s \right],$$

where the expectation is taken with respect to the process distribution induced by the policy. Ultimately, our objective is to find $\pi^* \in \arg\max_{\pi \in \Pi^{\mathrm{HR}}} v^\pi(s) =: v^*(s), \quad \forall s \in \mathcal{S}$. It is well-known that an optimal stationary deterministic policy exists in this standard MDP setting (Puterman, 2014).

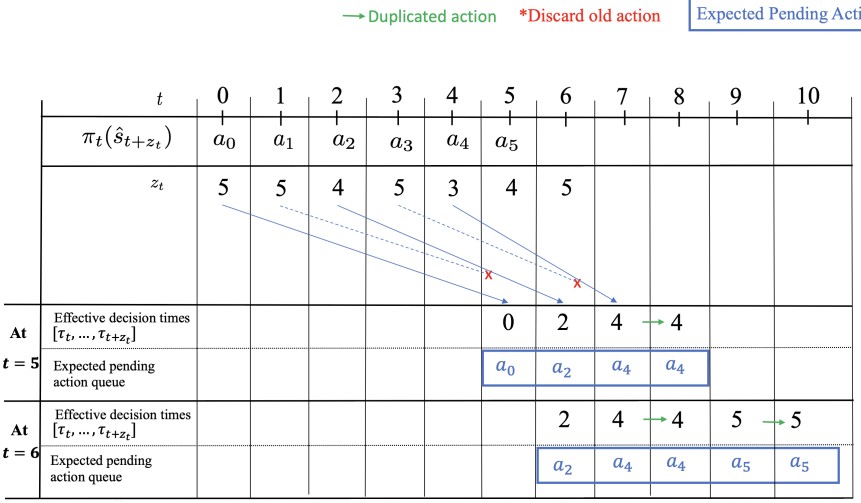

Figure 1: Pending queue resolution in a SED-MDP. The policy input $\hat{s}_{t+z_t}$ corresponds to the state inferred at $t$ by a forward model (see Section 5). For clarity, effective decision times are shown for $t \in \{5, 6\}$ only.

## 4 STOCHASTIC EXECUTION-DELAY MDP

In this section, we introduce stochastic execution-delay MDPs (SED-MDPs), which formalize stochastic delays and MDP dynamics as two distinct processes. We adopt the ED-MDP formulation of (Derman et al., 2021) that sidesteps state augmentation and extend it to the random delay case. With this perspective, we proceed in steps: (1) we define the SED-MDP framework with a suitable probability space; (2) we introduce the concept of *effective decision time*; (3) we prove the sufficiency of Markov policies to achieve optimal return in a SED-MDP.

A SED-MDP is a tuple $(\mathcal{S}, \mathcal{A}, P, r, \mu, \gamma, \zeta, \bar{a})$ such that $(\mathcal{S}, \mathcal{A}, P, r, \mu, \gamma)$ is an infinite-horizon MDP and $\zeta \in \Delta_{[M]}$ a distribution over possible delay values, where $[M] := \{0, 1, \cdots, M\}$. The last element $\bar{a} \in \mathcal{A}^M$ models a default queue of actions to be used in case of null assignment.[1] At each step $t$, a delay value $\tilde{z}_t$ is generated according to $\zeta$; the agent observes $\tilde{z}_t$ and a state $\tilde{s}_t$, prescribes an action $\tilde{a}_t$, receives a reward $r(\tilde{s}_t, \tilde{a}_t)$ and transits to $\tilde{s}_{t+1}$ according to $P(\cdot|\tilde{s}_t, \tilde{a}_t)$. Unlike the constant delay case, the action $\tilde{a}_t$ executed at time $t$ is sampled according to a policy that had been prescribed at a previous random time. This requires us to introduce the notion of effective decision time, which we shall describe in Section 4.1.

For any $\pi \in \Pi^{\text{HR}}$, the underlying probability space is $\Omega = (\mathcal{S} \times [M] \times \mathcal{A})^\infty$ which we assume to be equipped with a $\sigma$-algebra and a probability measure $\mathbb{P}^\pi$. Its elements are of the form $\omega = (s_0, z_0, a_0, s_1, z_1, a_1, \cdots)$. For all $t \in \mathbb{N}$, let the random variables $\tilde{s}_t : \Omega \to \mathcal{S}, \tilde{z}_t : \Omega \to [M]$ and $\tilde{a}_t : \Omega \to \mathcal{A}$ be respectively given by $\tilde{s}_t(\omega) = s_t, \tilde{z}_t(\omega) = z_t$ and $\tilde{a}_t(\omega) = a_t$. Clearly, $\tilde{a} := (\tilde{a}_t)_{t \in \mathbb{N}}$ and $\tilde{z} := (\tilde{z}_t)_{t \in \mathbb{N}}$ are dependent, as the distribution of $\tilde{a}_t$ depends on past realizations of $\tilde{z}_{:t}$ for all $t \in \mathbb{N}$. On the other hand, in this framework, the delay process $\tilde{z}$ is independent of the MDP dynamics while it is observed in real-time by the agent. The latter assumption is justified in applications such as communication networks, where delay can be effectively managed and controlled through adjustments in transmission power. We finally define the random variables of the history $(\tilde{h}_t)_{t \in \mathbb{N}}$ according to $\tilde{h}_0(\omega) = (s_0, z_0)$ and $\tilde{h}_t(\omega) = (s_0, z_0, a_0, \cdots, s_t, z_t), \quad \forall t \geq 1$.

### 4.1 EFFECTIVE DECISION TIME

To deal with random delays, we follow (Bouteiller et al., 2020) and assume that the SED-MDP actuates the most recent action that is available at that step. This implies that previously decided

---

[1]One may use a sequence of $M$ distributions instead of deterministically prescribed actions, but we restrict to actions for notation brevity.

actions can be overwritten or duplicated. Fig. 1 shows the queue dynamics resulting from successive realizations of the delay process. Concretely, at each step $t$, the delayed environment stores two distinct queues: one queue of $M$ past actions $(a_{t-M}, a_{t-M+1}, \cdots, a_{t-1})$ and one queue of $M$ past execution delays $(z_{t-M}, z_{t-M+1}, \cdots, z_{t-1})$. Then, we define the *effective decision time* at $t$ as:

$$\tau_t := \max_{t' \in [t-M:t]} \{t' + z_{t'} \leq t\}. \tag{1}$$

It is a stopping time in the sense that $\tau_t$ is $\sigma(\tilde{h}_t)$-measurable. Notably, it also determines the distribution that generated each executed action: given a policy $\pi = (\pi_t)_{t \geq 0}$, for any step $t$, the action at time $t$ is generated through the decision rule $\tilde{a}_t \sim \pi_{\tau_t}$. Intuitively, $\tau_t$ answers the question "What is the most recent action available for execution, given past and current delays"? Alternatively, we may define $e_t := t + z_t$ as the *earliest execution time* resulting from a decision taken at time $t$.

**Example.** Let $M = 5$ and assume that the following table gives the first realizations of decision rules and delays:

| $t$ | 0 | 1 | 2 | 3 | 4 |
|---|---|---|---|---|---|
| $z_t$ | 5 | 4 | 4 | 4 | 3 |
| $e_t$ | 5 | 5 | 6 | 7 | 7 |
| $a_t$ | $a_0$ | $a_1$ | $a_2$ | $a_3$ | $a_4$ |

As a result, the effective decision time at $t = 5$ is given by $\tau_5 = 1 \cdot \mathbb{1}_{\{z_5 > 0\}} + 5 \cdot \mathbb{1}_{\{z_5 = 0\}}$. From the agent's perspective, the pending queue at time $t = 5$ depends on the delay value $z_5$:

- If $z_5 = 5$, then the pending queue is $[a_1, a_2, a_4, a_4, a_4]$.
- If $z_5 = 4$, then the pending queue is $[a_1, a_2, a_4, a_4]$.
- If $z_5 = 3$, then the pending queue is $[a_1, a_2, a_4]$.
- If $z_5 = 2$, then the pending queue is $[a_1, a_2]$.
- If $z_5 = 1$, then the pending queue is $[a_1]$.
- If $z_5 = 0$, then the pending queue is $[]$.

## 4.2 SED-MDP PROCESS DISTRIBUTION

We now study the process distribution generated from any policy, which will lead us to establish the sufficiency of Markov policies for optimality. To begin with, the earliest time at which *some* action prescribed by the agent's policy is executed corresponds to $\min\{z_0, z_1 + 1, \ldots, M - 1 + z_{M-1}\}$ or equivalently, to $\min\{e_0, e_1, \ldots, e_{M-1}\}$. From there on, the SED-MDP performs a switch and uses the agent policy instead of the fixed default queue. Denoting $t_z := \min\{z_0, z_1 + 1, \ldots, M - 1 + z_{M-1}\}$, for any policy $\pi \in \Pi^{\mathrm{HR}}$, we have:

$$\mathbb{P}^\pi \left( \tilde{a}_t = a | \tilde{h}_t = (h_{t-1}, a_{t-1}, s_t, z_t) \right) = \begin{cases} \mathbb{1}_{\bar{a}_t}(a) & \text{if } t < \min\{z_0, z_1 + 1, \ldots, z_t + t\}, \\ \pi_{\tau_t}(h_t)(a) & \text{otherwise.} \end{cases} \tag{2}$$

We can now formulate the probability of a sampled trajectory in the following theorem. A proof is given in Appendix A.1.

**Theorem 4.1.** *For any policy $\pi := (\pi_t)_{t \in \mathbb{N}} \in \Pi^{\mathrm{HR}}$, the probability of observing history $h_t := (s_0, z_0, a_0, \cdots, a_{t-1}, s_t, z_t)$ is given by:*

$$\mathbb{P}^\pi(\tilde{s}_0 = s_0, \tilde{z}_0 = z_0, \tilde{a}_0 = a_0, \cdots, \tilde{a}_{t-1} = a_{t-1}, \tilde{s}_t = s_t, \tilde{z}_t = z_t)$$

$$= \mu(s_0) \left( \prod_{k=0}^{t_z - 1} \mathbb{1}_{\bar{a}_k}(a_k) P(s_{k+1}|s_k, a_k) \right) \left( \prod_{l=t_z}^{t-1} \pi_{\tau_l}(h_l)(a_l) P(s_{l+1}|s_l, a_l) \right) \left( \prod_{m=0}^{t} \zeta(z_m) \right).$$

Extending the result of (Derman et al., 2021), we establish that the process distribution generated from any history-dependent policy can equivalently be generated from a Markov policy. The proof is given in Appendix A.2.

**Theorem 4.2.** *Let* $\pi \in \Pi^{HR}$ *be a history-dependent policy. For all* $s_0 \in \mathcal{S}$, *there exists a Markov policy* $\pi' \in \Pi^{MR}$ *that yields the same process distribution as* $\pi$, *i.e.,* $\mathbb{P}^{\pi'}(\tilde{s}_{\tau_t} = s', \tilde{a}_t = a | \tilde{s}_0 = s_0, \tilde{z} = z) = \mathbb{P}^{\pi}(\tilde{s}_{\tau_t} = s', \tilde{a}_t = a | \tilde{s}_0 = s_0, \tilde{z} = z)$, *for all* $a \in \mathcal{A}, s' \in \mathcal{S}, t \geq 0$, *and* $\tilde{z} := (\tilde{z}_t)_{t \in \mathbb{N}}$ *the whole delay process.*

We can now devise policy training in the smaller class $\Pi^{MR}$ without impairing the agent's return. The algorithm we propose next does exactly this.

## 5 STOCHASTIC-DELAY EFFICIENTZERO

We introduce a novel algorithm designed to address the challenges posed by stochastic execution delay. Drawing inspiration from the recent achievements of EfficientZero on the Atari 100k benchmark (Ye et al., 2021), we use its architectural framework to infer future states with high accuracy. Its notable sample efficiency enables us to quickly train the forward model. We note that recent advancements in model-based approaches extend beyond lookahead search methods such as MuZero and EfficientZero. Depending on the task at hand, alternatives such as IRIS (Micheli et al., 2022), SimPLE (Kaiser et al., 2019), or DreamerV3 (Hafner et al., 2023) exhibit distinct characteristics that can also be considered. Our approach is adaptable and can be seamlessly integrated with any of these algorithms.

**Algorithm description.** DEZ is depicted in Fig. 2. It adapts EfficientZero (Ye et al., 2021) to act and learn in environments with stochastic delays. At any time $t$, we maintain two queues of length $M$. One is the action queue of previous realizations $M$ $[a_{t-M}, \ldots, a_{t-1}]$. The second is the delay value queue observed for these $M$ actions, $[z_{t-M}, \ldots, z_{t-1}]$. During inference, we observe state $s_t$ and delay $z_t$, and aim to estimate the future state $\hat{s}_{t+z_t}$. To do so, we take the expected pending actions denoted by $[\hat{a}_t, \ldots, \hat{a}_{t+z_t-1}]$ and successively apply the learned forward model: $\hat{s}_{t+1} = \mathcal{G}(s_t, \hat{a}_t), \ldots, \hat{s}_{t+z_t} = \mathcal{G}(\hat{s}_{t+z_t-1}, \hat{a}_{t+z_t-1})$, using notations from Ye et al. (2021). Finally, we perform an MCTS search to output $a_t := \pi_t(\hat{s}_{t+z_t})$ and add $a_t, z_t$ to the respective queues. On the other hand, the action executed in the delayed environment is $a_{\tau_t}$. Since no future action can overwrite the first expected action $\hat{a}_t$, we note that $a_{\tau_t} = \hat{a}_t$. The reward of the system is then stored to form a transition $(s_t, a_{\tau_t}, r_t)$.

Figure 2: Interaction diagram between DEZ and the delayed environment

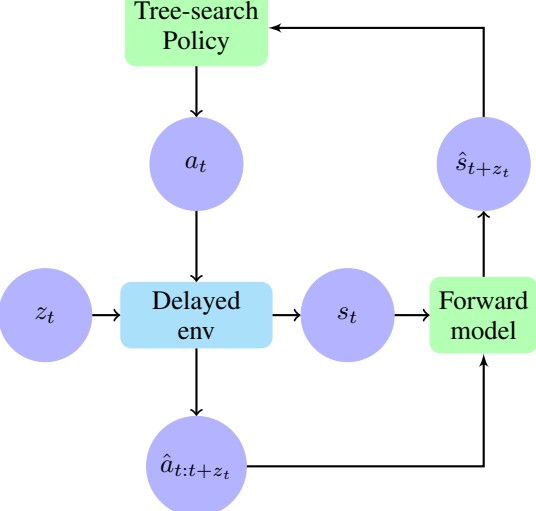

**Pending actions resolution.** Fig. 1 depicts the action and delay queues, together with the relation between effective decision times and pending actions. At each time step, the actual pending actions are calculated using the effective decision times from Eq. (1).

**Future state prediction.** To make accurate state predictions, we embrace the state representation network $s_t = \mathcal{H}(o_t)$ and the dynamics network $\hat{s}_{t+1} = \mathcal{G}(s_t, a_t)$ from EfficientZero (Ye et al., 2021). For simplicity, we do not distinguish between observation $o_t$ and state representation $s_t$ as do Ye et al. (2021). However, all parts of the network—-forward, dynamics, value, and policy—operate in the latent space. Moreover, the replay buffer contains transition samples with states $s_t, s_{t+1}$ in the same space. The pseudo-code in Algo. 1 depicts how self-play samples episodes in the stochastic delayed environment.

**Corrections in the replay buffer.** In the original version of EfficientZero, episode transitions $(s_t, a_t, r_t)$ are stored in the game histories that contain state, action, and reward per step. However, to insert the right transitions into the replay buffer, we post-process the game histories using all execution delays $\{z_t\}_{t=0}^{t=L}$ of the episode of length $L$, enabling us to compute effective decision times and store $(s_t, a_{\tau_t}, r_t)$ instead. More details about the episode processing can be found in Appx. B.

**Learning a policy in a delayed environment.** Action selection is carried out using the same MCTS approach as in EfficientZero. The output of this search denoted by $\pi_t(\hat{s}_{t+z_t})$ also serves as the policy target in the loss function. The primary alteration occurs in the policy loss, which takes the form:

$$\mathcal{L}_p = \mathcal{L}(\pi_t(\hat{s}_{t+z_t}), p_t(s_{t+z_t})).$$

where $p_t$ is the policy network at time $t$. Although the MCTS input relies on a predicted version of $s_{t+z_t}$, this loss remains justified due to the exceptional precision exhibited by the dynamics model within the latent space. Other loss components remain unchanged.

## 6 EXPERIMENTS

The comprehensive framework that we construct around EfficientZero accommodates both constant and stochastic execution delays. We consider the delay values $\{5, 15, 25\}$ to be either the constant value of delay or its maximal value when dealing with stochastic delays, as in (Derman et al., 2021). For both constant and stochastic delays, we refrain from random initialization of the initial action queue as in (Derman et al., 2021). Instead, our model determines the first $M$ actions. This is achieved through the iterative application of the forward and the policy networks. In practice, the agent observes the initial state $s_0$, infers the policy through $\bar{a}_0$, and predicts the subsequent state $\hat{s}_1 = \mathcal{G}(s_0, \bar{a}_0)$. It similarly infers $\hat{s}_2, \ldots, \hat{s}_M$ and selects actions $\bar{a}_2, \ldots, \bar{a}_{M-1}$ through this iterative process.

EfficientZero sampled 100K transitions, aligning with the Atari 100K benchmark. Although our approach significantly benefits from EfficientZero's sample efficiency, the presence of delay adds complexity to the learning process, requiring mildly more interactions – 130K in our case. Successfully tackling delays with such limited data is a non-trivial task.

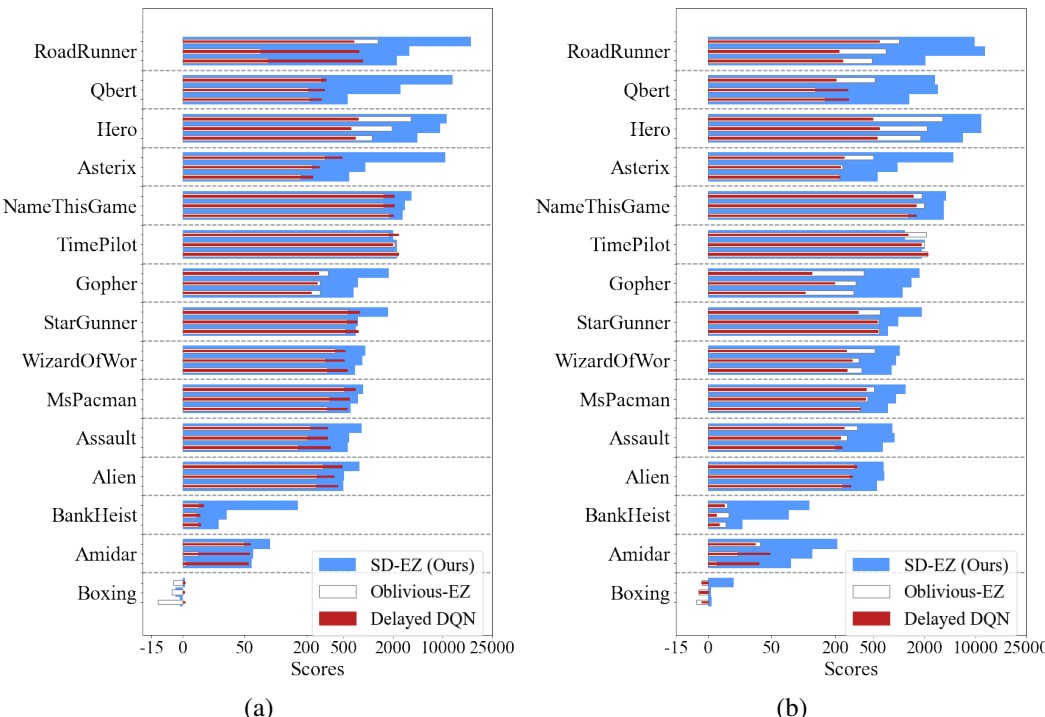

Figure 3: Average score on 15 Atari games and delays $M \in \{5, 15, 25\}$ over 32 test episodes per trained seed. Delays appear from low to high values for each game. (a) Constant delay value; (b) Stochastic delay value within $\{0, \cdots, M\}$.

To evaluate our theoretical findings, we subject DEZ to testing across 15 Atari games, each under both constant and stochastic delays. In either case, the delay value is revealed to the agent. We stress

that the baseline scores are not consistent for the two types of delays that we simulate. Indeed, on constant delays, Delayed-Q (Derman et al., 2021) tends to achieve higher scores than the oblivious version of EfficientZero, while Delayed-Q suffers more from stochastic delays than EfficientZero. As explained next, DEZ consistently achieves higher scores on both types of delays. A full summary of scores and standard deviations for our algorithm and baseline methods is presented in Appx. C.4, Tabs. 1 and 2.

## 6.1 CONSTANT DELAYS

We first analyze the scores produced by DEZ for constant delays and refer to Appx. C.1 for the convergence plots of all games tested.

Figure 3(a) shows that our method achieves the best average score in 39 of the 45 experiments. The oblivious version of EfficientZero confirms previously established theoretical findings (Derman et al., 2021, Prop. 5.2): stationary policies alone are insufficient to achieve the optimal value. Given that Oblivious EfficientZero lacks adaptability and exclusively generates stationary policies, it is not surprising that it struggles to learn effectively in most games.

Our second baseline reference is Delayed-Q (Derman et al., 2021), recognized as a state-of-the-art algorithm for addressing constant delays in Atari games. However, it should be noted that Delayed-Q is a DQN-based algorithm, and its original implementation entailed training with one million samples. Our observations reveal that within the constraints of the allotted number of samples, Delayed-Q struggles to achieve efficient learning. On the other hand, Delayed-Q benefits from its perfect forward model, provided by the environment itself. We observe that in 21 out of 45 experiments, Delayed-Q achieves an average score that is at least 85% of our method's result, even from the initial training iterations.

**Importance of the forward model.** Despite the dependence of Delayed-Q on the Atari environment as its forward model, this baseline exhibits significant vulnerabilities when faced with limited data. To address this challenge, we harness the representation learning capabilities of EfficientZero, allowing us to acquire state representations and dynamics through the utilization of its self-supervised rich reward signal, among other features. Our findings illustrate that when we effectively learn a precise prediction model, delays become more manageable. However, it is worth noting that increased randomness in the environment can result in larger compounding errors in subsequent forward models.

## 6.2 STOCHASTIC DELAYS

For a more realistic scenario, we also test our method on stochastic delays. For each Atari game, we use three stochastic delay settings. For each value $M \in \{5, 15, 25\}$, we use the following formulation of the delay $z_t$ at time $t$:

$$z_0 = M$$

$$z_{t>0} = \begin{cases} \min(z_t + 1, M) & \text{with probability } 0.2, \\ \max(z_t - 1, 0) & \text{with probability } 0.2, \\ z_t & \text{otherwise.} \end{cases}$$

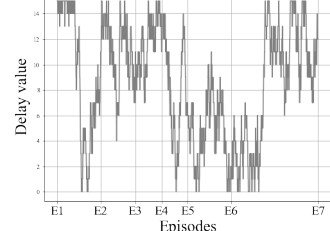

Note that the delay is never reinitialized to $M$ when the episode terminates. Instead, the delays remain the same for the beginning of the next episode. By doing that, we do not assume an initial delay value and cover a broader range of applications.

Figure 4: Random walk behavior of the stochastic delay across multiple episodes. No initial delay value is set at the start of the episode. Here, the maximal delay is 15.

We opt to avoid the augmentation approach in the baselines due to its inherent complexity. Incorporating the pending action queue into the state space was deemed infeasible due to its exponential growth, even in simple environments such as Maze (Derman et al., 2021). In our scenario, the length of the pending queue ranges from zero to $M$ in cases involving stochastic delays. Consequently, the size of the augmented state space becomes $|\mathcal{S}| \times |\mathcal{A}|^M$. As the value of $M$ increases, the order of magnitude of this quantity becomes overwhelming.

We report the average scores of DEZ and baselines for 15 Atari games in Figure 3(b), and refer to Appx. C.2 for the convergence plots. Similarly to the constant case, we observe an impressive achievement, even slightly better, highlighting the added resilience of our method to stochastic delays. In 42 of 45 experiments, our method prevails with the highest average score. DEZ is also trained to incorporate very large delay values for evaluation in extreme scenarios. The performance in such cases is predominantly influenced by the specific characteristics of the environment. For example, in Asterix, we observe that DEZ struggles to learn efficiently with delays of $M = 35$ and $M = 45$. Contrarily, in Hero, our algorithm demonstrates comparable learning performance even with the largest delay value.

## 7 DISCUSSION

In this work, we formalized stochastic execution delay in MDPs without resorting to state augmentation. Although its analysis is more engaged, this setting leads to an intriguing insight: as in the constant delay case, we can restrict the policy search to the class of Markov policies and still reach optimal performance in the stochastic delay case. We introduced DEZ, a model-based approach that achieves state-of-the-art performance for delayed environments, thanks to an accurately learned world model. A natural extension of our approach would consider predicting the next state distribution instead of point prediction $\hat{s}_{t+1}$ to train a more robust world model and mitigate uncertainty in more stochastic environments.

DEZ heavily relies on a learned world model to infer future states. Devising a model-free method that addresses delayed execution while avoiding embedding remains an open question. DEZ also assumes delay values are observed in real-time, which enabled us to backpropagate the forward model the corresponding number of times. Ridding of this constraint would add another layer of difficulty because the decision time that generated each action would be ignored. For deterministic execution delay, we may estimate its value. For stochastic delays, we could adopt a robust approach to roll out multiple realizations of delays and act according to a worst-case criterion.

Useful improvements may also be made to support continuous delays and remove drop and duplication of actions. In such a case, we may plan and derive continuous actions. Another modeling assumption that we made was that the delay process is independent of the MDP environment. Alternatively, one may study dependent delay processes, e.g., delay values that depend on the agent's current state or decision rule. Such extension is particularly meaningful in autonomous driving where the agent state must be interpreted from pictures of various information complexities.

## 8 REPRODUCIBILITY

To further advance the cause of reproducibility in the field of Reinforcement Learning (RL), we are committed to transparency and rigor throughout our research process. We outline the steps taken to facilitate reproducibility: (1) Detailed Methodology: Our commitment to transparency begins with a description of our methodology. We provide clear explanations of the experimental setup, algorithmic choices, and concepts introduced to ensure that fellow researchers can replicate our work. (2) Formal proofs are presented in the appendix. (3) Open implementation: To encourage exploration and research on delay-related challenges in RL, we include the code as part of the supplementary material. We include a README file with instructions on how to run both training and evaluation. Our repository is a fork of EfficientZero (Ye et al., 2021) with the default parameters taken from the original paper. (4) Acknowledging Distributed Complexity: It is important to note that, while we strive for reproducibility, the inherent complexity of distributed architectures, such as EfficientZero, presents certain challenges. Controlling the order of execution through code can be intricate, and we acknowledge that achieving exact-result replication in these systems may pose difficulties. In summary, our dedication to reproducibility is manifested through transparent methodologies, rigorous formal proofs, and an openly accessible implementation. Although we recognize the complexities of distributed systems, we believe that our contributions provide valuable insight into delay-related issues in RL, inviting further collaboration within the research community.

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

APPENDIX

# A  A FORMULATION FOR STOCHASTIC DELAYS

## A.1  PROOF OF THEOREM 4.1

*Proof.* By definition of conditional probability, for all measurable sets $A_1, \cdots, A_n \in \mathcal{B}(\Omega)$, we have

$$\mathbb{P}^\pi(\cap_{i=0}^n A_i) = \left( \prod_{i=0}^{n-1} \mathbb{P}^\pi(A_i | \cap_{j=i+1}^n A_j) \right) \mathbb{P}^\pi(A_n). \tag{3}$$

Applying Eq. (3) to $n = 2t + 1$ on the following events:

$$A_{2t+1} := \{\tilde{s}_0 = s_0\}$$
$$A_{2t} := \{\tilde{z}_0 = z_0, \tilde{a}_0 = a_0\}$$
$$\vdots$$
$$A_2 := \{\tilde{z}_{t-1} = z_{t-1}, \tilde{a}_{t-1} = a_{t-1}\}$$
$$A_1 := \{\tilde{s}_t = s_t\},$$
$$A_0 := \{\tilde{z}_t = z_t\}$$

we obtain that

$$\mathbb{P}^\pi(\tilde{s}_0 = s_0, \tilde{z}_0 = z_0, \tilde{a}_0 = a_0, \cdots, \tilde{a}_{t-1} = a_{t-1}, \tilde{s}_t = s_t, \tilde{z}_t = z_t)$$

$$= \mathbb{P}^\pi(\tilde{s}_0 = s_0) \prod_{i=0}^{t-1} \mathbb{P}^\pi(\tilde{z}_i = z_i, \tilde{a}_i = a_i | \tilde{s}_0 = s_0, \tilde{z}_0 = z_0, \tilde{a}_0 = a_0, \cdots, \tilde{s}_i = s_i)$$

$$\mathbb{P}^\pi(\tilde{s}_{i+1} = s_{i+1} | \tilde{s}_0 = s_0, \tilde{z}_0 = z_0, \tilde{a}_0 = a_0, \cdots, \tilde{z}_i = z_i, \tilde{a}_i = a_i)$$
$$\cdot \mathbb{P}^\pi(\tilde{z}_t = z_t | \tilde{s}_0 = s_0, \tilde{z}_0 = z_0, \tilde{a}_0 = a_0, \cdots, \tilde{z}_{t-1} = z_{t-1}, \tilde{a}_{t-1} = a_{t-1}, \tilde{s}_t = s_t)$$

$$\overset{(1)}{=} \mathbb{P}^\pi(\tilde{s}_0 = s_0) \prod_{i=0}^{t-1} \mathbb{P}^\pi(\tilde{z}_i = z_i, \tilde{a}_i = a_i | \tilde{s}_0 = s_0, \tilde{z}_0 = z_0, \tilde{a}_0 = a_0, \cdots, \tilde{s}_i = s_i)$$

$$\mathbb{P}^\pi(\tilde{s}_{i+1} = s_{i+1} | \tilde{s}_0 = s_0, \tilde{z}_0 = z_0, \tilde{a}_0 = a_0, \cdots, \tilde{z}_i = z_i, \tilde{a}_i = a_i) \mathbb{P}^\pi(\tilde{z}_t = z_t)$$

$$= \mathbb{P}^\pi(\tilde{s}_0 = s_0) \prod_{i=0}^{t-1} \mathbb{P}^\pi(\tilde{z}_i = z_i, \tilde{a}_i = a_i | \tilde{s}_0 = s_0, \tilde{z}_0 = z_0, \tilde{a}_0 = a_0, \cdots, \tilde{s}_i = s_i)$$

$$\mathbb{P}^\pi(\tilde{s}_{i+1} = s_{i+1} | \tilde{h}_i = (h_{i-1}, a_{i-1}, z_i, s_i), \tilde{a}_i = a_i) \mathbb{P}^\pi(\tilde{z}_t = z_t)$$

$$= \mathbb{P}^\pi(\tilde{s}_0 = s_0) \prod_{i=0}^{t-1} \mathbb{P}^\pi(\tilde{z}_i = z_i, \tilde{a}_i = a_i | \tilde{s}_0 = s_0, \tilde{z}_0 = z_0, \tilde{a}_0 = a_0, \cdots, \tilde{s}_i = s_i) P(s_{i+1}|s_i, a_i)\zeta(z_t)$$

$$= \mathbb{P}^\pi(\tilde{s}_0 = s_0) \prod_{i=0}^{t-1} \mathbb{P}^\pi(\tilde{a}_i = a_i | \tilde{h}_i = h_i) \mathbb{P}^\pi(\tilde{z}_i = z_i | \tilde{s}_0 = s_0, \tilde{z}_0 = z_0, \tilde{a}_0 = a_0, \cdots, \tilde{s}_i = s_i) P(s_{i+1}|s_i, a_i)\zeta(z_t)$$

$$= \mathbb{P}^\pi(\tilde{s}_0 = s_0) \prod_{i=0}^{t-1} \mathbb{P}^\pi(\tilde{a}_i = a_i | \tilde{h}_i = h_i)\zeta(z_i) P(s_{i+1}|s_i, a_i)\zeta(z_t)$$

$$= \mu(s_0) \left( \prod_{k=0}^{t_z-1} \mathbb{1}_{\bar{a}_k}(a_k) P(s_{k+1}|s_k, a_k) \right) \left( \prod_{l=t_z}^{t-1} \pi_{\tau_l}(h_l)(a_l) P(s_{l+1}|s_l, a_l) \right) \zeta(z_t)$$

where equality (1) comes from the fact that the delay distribution at $i+1$ is independent of the state $s_{i+1}$ observed then. By Eq. (2),

$$\mathbb{P}^\pi(\tilde{s}_0 = s_0, \tilde{z}_0 = z_0, \tilde{a}_0 = a_0, \cdots, \tilde{a}_{t-1} = a_{t-1}, \tilde{s}_t = s_t, \tilde{z}_t = z_t)$$

$$= \mathbb{P}^\pi(\tilde{s}_0 = s_0) \left( \prod_{i=0}^{t} \zeta(z_i) \right) \prod_{i=0}^{t_z - 1} \mathbb{P}^\pi(\tilde{a}_i = a_i | \tilde{h}_i = h_i) \mathbb{P}^\pi(\tilde{s}_{i+1} = s_{i+1} | \tilde{h}_i = (h_{i-1}, a_{i-1}, s_i, z_i), \tilde{a}_i = a_i)$$

$$\prod_{k=t_z}^{t-1} \mathbb{P}^\pi(\tilde{a}_k = a_k | \tilde{h}_k = h_k) \mathbb{P}^\pi(\tilde{s}_{k+1} = s_{k+1} | \tilde{h}_k = (h_{k-1}, a_{k-1}, s_k, z_k), \tilde{a}_k = a_k)$$

$$= \mu(s_0) \left( \prod_{k=0}^{t_z - 1} \mathbb{1}_{\bar{a}_k}(a_k) P(s_{k+1} | s_k, a_k) \right) \left( \prod_{l=t_z}^{t-1} \pi_{\tau_l}(h_l)(a_l) P(s_{l+1} | s_l, a_l) \right) \left( \prod_{m=0}^{t} \zeta(z_m) \right),$$

where the first product is empty if and only if $z_0 = 0$, i.e., the decision taken at time $t = 0$ is immediately executed. □

### A.2 Proof of Theorem 4.2

We first establish the following lemma, which will be used in the theorem's proof.

**Lemma A.1.** *For all $t > 0$ and $z_t > 0$,*

$$\mathbb{P}^\pi(\tilde{s}_t = s' | \tilde{a}_t = a', \tilde{s}_{t-1} = s, \tilde{a}_{t-1} = a) = \mathbb{P}^\pi(\tilde{s}_t = s' | \tilde{s}_{t-1} = s, \tilde{a}_{t-1} = a) \tag{4}$$

*Proof.* Since $z_t > 0$ by assumption, the state variable $\tilde{s}_t$ is independent of $\tilde{a}_t$, so that: $\mathbb{P}^\pi(\tilde{s}_t = s' | \tilde{a}_t = a', \tilde{s}_{t-1} = s, \tilde{a}_{t-1} = a) = P(s' | s, a) = \mathbb{P}^\pi(\tilde{s}_t = s' | \tilde{s}_{t-1} = s, \tilde{a}_{t-1} = a)$.

□

**Theorem.** *Let $\pi \in \Pi^{HR}$ be a history dependent policy. For all $s_0 \in \mathcal{S}$, there exists a Markov policy $\pi' \in \Pi^{MR}$ that yields the same process distribution as $\pi$, i.e.,*

$$\mathbb{P}^{\pi'}(\tilde{s}_{\tau_t} = s', \tilde{a}_t = a | \tilde{s}_0 = s_0, \tilde{z} = z) = \mathbb{P}^\pi(\tilde{s}_{\tau_t} = s', \tilde{a}_t = a | \tilde{s}_0 = s_0, \tilde{z} = z), \tag{5}$$

*for all $a \in \mathcal{A}, s' \in \mathcal{S}, t \geq 0$, and $\tilde{z} := (\tilde{z}_t)_{t \in \mathbb{N}}$ the whole delay process.*

*Proof.* If $M = 0$, the result holds true by standard RL theory (Puterman, 2014)[Thm 5.5.1]. Assume now that $M > 0$, fix $s \in \mathcal{S}$ and $z = (z_t)_{t \in \mathbb{N}}$. The variable $z$ is exogenous, and we shall consider the corresponding sequence of effective decision times $(\tau_t)_{t \in \mathbb{N}}$. We construct a Markov policy $\pi'$ for these times only while for other times, $\pi'$ can be anything else. Let thus $\pi' := (\pi'_{\tau_t})_{t \in \mathbb{N}}$ with $\pi'_{\tau_t} : \{s\} \to \Delta_\mathcal{A}$ defined as

$$\pi'_{\tau_t}(s')(a) := \begin{cases} \mathbb{P}^\pi(\tilde{a}_t = a | \tilde{s}_{\tau_t} = s', \tilde{s}_0 = s, \tilde{z} = z) & \text{if } t \geq t_z, \\ \mathbb{1}_{\bar{a}_t}(a) & \text{otherwise,} \end{cases} \quad \forall s' \in \mathcal{S}, a \in \mathcal{A}. \tag{6}$$

Recall the definition of the time the SED-MDP performs a switch and uses the agent's policy $t_z := \min\{z_0, z_1 + 1, \ldots, M - 1 + z_{M-1}\}$. For the policy $\pi'$ defined in Eq. (6), we prove the theorem by induction on $t \geq t_z$, since for $t < t_z$, the decision rule being applied is $\mathbb{1}_{\bar{a}_t}$ regardless of the policy. For $t = t_z$, by Thm. 4.1, we have:

$$\mathbb{P}^\pi(\tilde{s}_{\tau_{t_z}} = s', \tilde{a}_{t_z} = a | \tilde{s}_0 = s_0) = \frac{\mathbb{P}^\pi(\tilde{s}_{\tau_{t_z}} = s', \tilde{a}_{t_z} = a, \tilde{s}_0 = s_0)}{\mathbb{P}^\pi(\tilde{s}_0 = s_0)}$$

$$= \frac{1}{\mathbb{P}^\pi(\tilde{s}_0 = s_0)} \sum_{\substack{s_1, \cdots, s_{t_z - 1}, \\ a_0, \cdots, a_{t_z - 1}}} \mathbb{P}^\pi(\tilde{s}_{\tau_{t_z}} = s', \tilde{a}_{t_z} = a, \tilde{a}_{t_z - 1} = a_{t_z - 1}, \cdots, \tilde{a}_0 = a_0, \tilde{s}_0 = s_0)$$

$$= \sum_{\substack{s_1, \cdots, s_{t_z - 1}, \\ a_0, \cdots, a_{t_z - 1}}} \prod_{k=0}^{t_z - 1} \mathbb{1}_{\bar{a}_k}(a_k) P(s_{k+1} | s_k, a_k)$$

We observe that the expression does not depend on the prescribed policy, so it equals $\mathbb{P}^{\pi'}(\tilde{s}_{\tau_{t_z}} = s', \tilde{a}_{t_z} = a | \tilde{s}_0 = s_0)$ and the induction base holds.

Assume that Eq. (5) holds up until $t = n - 1$, with $t \geq t_z$. Let denote $t' = n$, we aim to prove that

$$\mathbb{P}^{\pi}(\tilde{s}_{t'} = s' | \tilde{s}_0 = s) = \mathbb{P}^{\pi'}(\tilde{s}_{t'} = s' | \tilde{s}_0 = s). \tag{7}$$

Then, we can write

$$\mathbb{P}^{\pi}(\tilde{s}_{t'} = s' | \tilde{s}_0 = s)$$
$$= \sum_{\substack{s_t \in \mathcal{S}, \\ a_t \in \mathcal{A}}} \mathbb{P}^{\pi}(\tilde{s}_{t'} = s', \tilde{s}_t = s_t, \tilde{a}_t = a_t | \tilde{s}_0 = s)$$
$$= \sum_{\substack{s_t \in \mathcal{S}, \\ a_t \in \mathcal{A}}} \mathbb{P}^{\pi}(\tilde{s}_{t'} = s' | \tilde{a}_t = a_t, \tilde{s}_t = s_t, \tilde{s}_0 = s)$$
$$\mathbb{P}^{\pi}(\tilde{a}_t = a_t | \tilde{s}_t = s_t, \tilde{s}_0 = s) \mathbb{P}^{\pi}(\tilde{s}_t = s_t | \tilde{s}_0 = s)$$
$$= \sum_{\substack{s_t \in \mathcal{S}, \\ a_t \in \mathcal{A}}} P(s' | s_t, a_t) \mathbb{P}^{\pi}(\tilde{a}_t = a_t | \tilde{s}_t = s_t, \tilde{s}_0 = s) \mathbb{P}^{\pi}(\tilde{s}_t = s_t | \tilde{s}_0 = s).$$

By Eq. 2, $\tilde{a}_t$ only depends on state-action sequences up to $\tau_t$. Let's differentiate on the value of $z_t$. If $z_t = 0$, then $\tau_t = t$ and by construction of $\pi'$ we can write:

$$\mathbb{P}^{\pi}(\tilde{a}_t = a_t | \tilde{s}_t = s_t, \tilde{s}_0 = s) = \mathbb{P}^{\pi}(\tilde{a}_t = a_t | \tilde{s}_{\tau_t} = s_t, \tilde{s}_0 = s) = \mathbb{P}^{\pi'}(\tilde{a}_t = a_t | \tilde{s}_{\tau_t} = s_t, \tilde{s}_0 = s).$$

$$\Rightarrow \mathbb{P}^{\pi}(\tilde{s}_{t'} = s' | \tilde{s}_0 = s) = \sum_{\substack{s_t \in \mathcal{S}, \\ a_t \in \mathcal{A}}} P(s' | s_t, a_t) \mathbb{P}^{\pi'}(\tilde{a}_t = a_t | \tilde{s}_{\tau_t} = s_t, \tilde{s}_0 = s) \mathbb{P}^{\pi}(\tilde{s}_t = s_t | \tilde{s}_0 = s).$$

If $z_t > 0$, then $\tau_t < t$ and we can write: $\mathbb{P}^{\pi}(\tilde{a}_t = a_t | \tilde{s}_t = s_t, \tilde{s}_0 = s) = \mathbb{P}^{\pi}(\tilde{a}_t = a_t | \tilde{s}_0 = s)$

$$\Rightarrow \mathbb{P}^{\pi}(\tilde{s}_{t'} = s' | \tilde{s}_0 = s) = \sum_{\substack{s_t \in \mathcal{S}, \\ a_t \in \mathcal{A}}} P(s' | s_t, a_t) \mathbb{P}^{\pi}(\tilde{a}_t = a_t | \tilde{s}_0 = s) \mathbb{P}^{\pi}(\tilde{s}_t = s_t | \tilde{s}_0 = s).$$

Since $t = n - 1$, by the induction hypothesis we can rewrite

$$\mathbb{P}^{\pi}(\tilde{a}_t = a_t | \tilde{s}_0 = s) = \sum_{s_{\tau_t} \in \mathcal{S}} \mathbb{P}^{\pi}(\tilde{a}_t = a_t, \tilde{s}_{\tau_t} = s_{\tau_t} | \tilde{s}_0 = s)$$
$$= \sum_{s_{\tau_t} \in \mathcal{S}} \mathbb{P}^{\pi'}(\tilde{a}_t = a_t, \tilde{s}_{\tau_t} = s_{\tau_t} | \tilde{s}_0 = s)$$
$$= \mathbb{P}^{\pi'}(\tilde{a}_t = a_t | \tilde{s}_0 = s)$$
$$\Rightarrow \mathbb{P}^{\pi}(\tilde{s}_{t'} = s' | \tilde{s}_0 = s) = \sum_{\substack{s_t \in \mathcal{S}, \\ a_t \in \mathcal{A}}} P(s' | s_t, a_t) \mathbb{P}^{\pi'}(\tilde{a}_t = a_t | \tilde{s}_0 = s) \mathbb{P}^{\pi}(\tilde{s}_t = s_t | \tilde{s}_0 = s).$$

In the two cases, we now study the last term in the above equations, $\mathbb{P}^{\pi}(\tilde{s}_t = s_t | \tilde{s}_0)$. We have

$$\mathbb{P}^{\pi}(\tilde{s}_t = s_t | \tilde{s}_0 = s)$$

$$= \sum_{\substack{s_{t-1}, \cdots, s_{t_z} \in \mathcal{S} \\ a_{t-1}, \cdots, a_{t_z} \in \mathcal{A}}} \mathbb{P}^{\pi}(\tilde{s}_t = s_t, \tilde{s}_{t-1} = s_{t-1}, \tilde{a}_{t-1} = a_{t-1}, \cdots, \tilde{s}_{t_z} = s_{t_z}, \tilde{a}_{t_z} = a_{t_z} | \tilde{s}_0 = s)$$

$$= \sum_{\substack{s_{t-1}, \cdots, s_{t_z} \in \mathcal{S} \\ a_{t-1}, \cdots, a_{t_z} \in \mathcal{A}}} \mathbb{P}^{\pi}(\tilde{s}_t = s_t | \tilde{s}_{t-1} = s_{t-1}, \tilde{a}_{t-1} = a_{t-1}, \cdots, \tilde{s}_{t_z} = s_{t_z}, \tilde{a}_{t_z} = a_{t_z}, \tilde{s}_0 = s)$$

$$\mathbb{P}^{\pi}(\tilde{s}_{t-1} = s_{t-1}, \tilde{a}_{t-1} = a_{t-1}, \cdots, \tilde{s}_{t_z} = s_{t_z}, \tilde{a}_{t_z} = a_{t_z} | \tilde{s}_0 = s)$$

$$= \sum_{\substack{s_{t-1}, \cdots, s_{t_z} \in \mathcal{S} \\ a_{t-1}, \cdots, a_{t_z} \in \mathcal{A}}} P(s_t | s_{t-1}, a_{t-1})$$

$$\mathbb{P}^{\pi}(\tilde{s}_{t-1} = s_{t-1}, \tilde{a}_{t-1} = a_{t-1}, \cdots, \tilde{s}_{t_z} = s_{t_z}, \tilde{a}_{t_z} = a_{t_z} | \tilde{s}_0 = s)$$

$$= \sum_{\substack{s_{t-1}, \cdots, s_{t_z} \in \mathcal{S} \\ a_{t-1}, \cdots, a_{t_z} \in \mathcal{A}}} P(s_t | s_{t-1}, a_{t-1})$$

$$\mathbb{P}^{\pi}(\tilde{s}_{t-1} = s_{t-1} | \tilde{a}_{t-1} = a_{t-1}, \tilde{s}_{t-2} = s_{t-2}, \tilde{a}_{t-2} = a_{t-2}, \cdots, \tilde{s}_{t_z} = s_{t_z}, \tilde{a}_{t_z} = a_{t_z}, \tilde{s}_0 = s)$$

$$\mathbb{P}^{\pi}(\tilde{a}_{t-1} = a_{t-1}, \tilde{s}_{t-2} = s_{t-2}, \tilde{a}_{t-2} = a_{t-2}, \cdots, \tilde{s}_{t_z} = s_{t_z}, \tilde{a}_{t_z} = a_{t_z} | \tilde{s}_0 = s)$$

$$\overset{\text{Lemma A.1}}{=} \sum_{\substack{s_{t-1}, \cdots, s_{t_z} \in \mathcal{S} \\ a_{t-1}, \cdots, a_{t_z} \in \mathcal{A}}} P(s_t | s_{t-1}, a_{t-1}) P(s_{t-1} | s_{t-2}, a_{t-2})$$

$$\mathbb{P}^{\pi}(\tilde{a}_{t-1} = a_{t-1}, \tilde{s}_{t-2} = s_{t-2}, \tilde{a}_{t-2} = a_{t-2}, \cdots, \tilde{s}_{t_z} = s_{t_z}, \tilde{a}_{t_z} = a_{t_z} | \tilde{s}_0 = s)$$

$$= \sum_{\substack{s_{t-1}, \cdots, s_{t_z} \in \mathcal{S} \\ a_{t-1}, \cdots, a_{t_z} \in \mathcal{A}}} P(s_t | s_{t-1}, a_{t-1}) P(s_{t-1} | s_{t-2}, a_{t-2})$$

$$\mathbb{P}^{\pi}(\tilde{a}_{t-1} = a_{t-1} | \tilde{s}_{t-2} = s_{t-2}, \tilde{a}_{t-2} = a_{t-2} \cdots, \tilde{s}_{t_z} = s_{t_z}, \tilde{a}_{t_z} = a_{t_z} | \tilde{s}_0 = s)$$

$$\mathbb{P}^{\pi}(\tilde{s}_{t-2} = s_{t-2}, \tilde{a}_{t-2} = a_{t-2} \cdots, \tilde{s}_{t_z} = s_{t_z}, \tilde{a}_{t_z} = a_{t_z} | \tilde{s}_0 = s)$$

$$=$$

$$\vdots$$

$$= \sum_{\substack{s_{t-1}, \cdots, s_{t_z} \in \mathcal{S} \\ a_{t-1}, \cdots, a_{t_z} \in \mathcal{A}}} \left( \prod_{i=0}^{t-t_z} P(s_{t-i} | s_{t-i-1}, a_{t-i-1}) \right)$$

$$\left( \prod_{j=1}^{t-t_z-1} \mathbb{P}^{\pi}(\tilde{a}_{t-j} = a_{t-j} | \tilde{s}_{t-j-1} = s_{t-j-1}, \tilde{a}_{t-j-1} = a_{t-j-1}, \cdots, \tilde{s}_{t_z} = s_{t_z}, \tilde{a}_{t_z} = a_{t_z}, \tilde{s}_0 = s) \right)$$

$$\mathbb{P}^{\pi}(\tilde{s}_{t_z} = s_{t_z} | \tilde{a}_{t_z} = a_{t_z}, \tilde{s}_0 = s) \mathbb{P}^{\pi}(\tilde{a}_{t_z} = a_{t_z} | \tilde{s}_0 = s)$$

$$\overset{(1)}{=} \sum_{\substack{s_{t-1}, \cdots, s_{t_z} \in \mathcal{S} \\ a_{t-1}, \cdots, a_{t_z} \in \mathcal{A}}} \left( \prod_{i=0}^{t-t_z} P(s_{t-i} | s_{t-i-1}, a_{t-i-1}) \right) \left( \prod_{j=1}^{t-t_z-1} \mathbb{P}^{\pi}(\tilde{a}_{t-j} = a_{t-j} | \tilde{s}_{\tau_{t-j}} = s_{\tau_{t-j}}) \right)$$

$$\mathbb{P}^{\pi}(\tilde{s}_{t_z} = s_{t_z} | \tilde{a}_{t_z} = a_{t_z}, \tilde{s}_0 = s) \mathbb{P}^{\pi}(\tilde{a}_{t_z} = a_{t_z} | \tilde{s}_0 = s)$$

We continue and write

$$\overset{(2)}{=} \sum_{\substack{s_{t-1}, \cdots, s_{t_z} \in \mathcal{S} \\ a_{t-1}, \cdots, a_{t_z} \in \mathcal{A}}} \left( \prod_{i=0}^{t-t_z} P(s_{t-i} | s_{t-i-1}, a_{t-i-1}) \right) \left( \prod_{j=1}^{t-t_z-1} q_{d'_{\tau_{t-j}}(s_{\tau_{t-j}})}(a_{t-j}) \right)$$

$$\mathbb{P}^{\pi}(\tilde{s}_{t_z} = s_{t_z} | \tilde{a}_{t_z} = a_{t_z}, \tilde{s}_0 = s) \mathbb{P}^{\pi}(\tilde{a}_{t_z} = a_{t_z} | \tilde{s}_0 = s)$$

$$\overset{(3)}{=} \sum_{\substack{s_{t-1}, \cdots, s_{t_z} \in \mathcal{S} \\ a_{t-1}, \cdots, a_{t_z} \in \mathcal{A}}} \left( \prod_{i=0}^{t-t_z} P(s_{t-i} | s_{t-i-1}, a_{t-i-1}) \right) \left( \prod_{j=1}^{t-t_z-1} q_{d'_{\tau_{t-j}}(s_{\tau_{t-j}})}(a_{t-j}) \right)$$

$$\mathbb{P}^{\pi}(\tilde{s}_{t_z} = s_{t_z} | \tilde{a}_{t_z} = a_{t_z}, \tilde{s}_0 = s) \sum_{s'_{\tau_{t_z}} \in \mathcal{S}} \left( q_{d'_{\tau_{t_z}}(s_{\tau_{t_z}})}(a_{t_z}) \right)$$

In $(2)$ and $(3)$ we used equation 2. We now analyze the last term implying on $\pi$ and show that it is not policy dependant:

$$\mathbb{P}^\pi(\tilde{s}_{t_z} = s_{t_z}|\tilde{a}_{t_z} = a_{t_z}, \tilde{s}_0 = s) = \mathbb{P}^\pi(\tilde{s}_{t_z} = s_{t_z}|\tilde{s}_0 = s)$$

$$= \sum_{\substack{s_{t_z-1},\cdots,s_1 \in \mathcal{S} \\ a_{t_z-1},\cdots,a_0 \in \mathcal{A}}} \mathbb{P}^\pi(\tilde{s}_{t_z} = s_{t_z}, \tilde{s}_{t_z-1} = s_{t_z-1}, \tilde{a}_{t_z-1} = a_{t_z-1}, \cdots, \tilde{s}_1 = s_1, \tilde{a}_1 = a_1, \tilde{a}_0 = a_0|\tilde{s}_0 = s)$$

$$= \sum_{\substack{s_{t_z-1},\cdots,s_1 \in \mathcal{S} \\ a_{t_z-1},\cdots,a_0 \in \mathcal{A}}} \frac{1}{\mathbb{P}^\pi(\tilde{s}_0 = s)} \mathbb{P}^\pi(\tilde{s}_{t_z} = s_{t_z}, \tilde{s}_{t_z-1} = s_{t_z-1}, \tilde{a}_{t_z-1} = a_{t_z-1}, \cdots, \tilde{s}_1 = s_1, \tilde{a}_1 = a_1, \tilde{a}_0 = a_0, \tilde{s}_0 = s)$$

$$\overset{(4)}{=} \sum_{\substack{s_{t_z-1},\cdots,s_1 \in \mathcal{S} \\ a_{t_z-1},\cdots,a_0 \in \mathcal{A}}} \frac{1}{\mu(s)} \mu(s) \left(\prod_{i=0}^{t_z-1} P(s_{i+1}|s_i, a_i)\mathbb{1}_{\bar{a}_i}(a_i)\right) = \sum_{\substack{s_{t_z-1},\cdots,s_1 \in \mathcal{S} \\ a_{t_z-1},\cdots,a_0 \in \mathcal{A}}} \left(\prod_{i=0}^{t_z-1} P(s_{i+1}|s_i, a_i)\mathbb{1}_{\bar{a}_i}(a_i)\right),$$

where $(4)$ results from Th. 4.1.

Thus, if we decompose $\mathbb{P}^{\pi'}(\tilde{s}_{t'} = s'|\tilde{s}_0 = s)$ according to the exact same derivation as we did for $\mathbb{P}^\pi(\tilde{s}_{t'} = s'|\tilde{s}_0 = s)$, we obtain that at $t' = n$:

$$\mathbb{P}^\pi(\tilde{s}_{t'} = s'|\tilde{s}_0 = s) = \mathbb{P}^{\pi'}(\tilde{s}_{t'} = s'|\tilde{s}_0 = s). \tag{8}$$

As a preceding step in the induction process, this results holds at $\tau_{t'} \le t' = n$:

$$\mathbb{P}^\pi(\tilde{s}_{\tau_{t'}} = s'|\tilde{s}_0 = s) = \mathbb{P}^{\pi'}(\tilde{s}_{\tau_{t'}} = s'|\tilde{s}_0 = s). \tag{9}$$

As a result, at $t' = n$ we have

$$\mathbb{P}^{\pi'}(\tilde{s}_{\tau_{t'}} = s', \tilde{a}_{t'} = a|\tilde{s}_0 = s) = \mathbb{P}^{\pi'}(\tilde{a}_{t'} = a|\tilde{s}_{\tau_{t'}} = s', \tilde{s}_0 = s)\mathbb{P}^{\pi'}(\tilde{s}_{\tau_{t'}} = s'|\tilde{s}_0 = s)$$

$$\overset{(a)}{=} \mathbb{P}^{\pi'}(\tilde{a}_{t'} = a|\tilde{s}_{\tau_{t'}} = s', \tilde{s}_0 = s)\mathbb{P}^\pi(\tilde{s}_{\tau_{t'}} = s'|\tilde{s}_0 = s)$$

$$\overset{(b)}{=} \mathbb{P}^\pi(\tilde{a}_{t'} = a|\tilde{s}_{\tau_{t'}} = s', \tilde{s}_0 = s)\mathbb{P}^\pi(\tilde{s}_{\tau_{t'}} = s'|\tilde{s}_0 = s)$$

$$= \mathbb{P}^\pi(\tilde{s}_{\tau_{t'}} = s', \tilde{a}_{t'} = a|\tilde{s}_0 = s),$$

where $(a)$ follows from Eq. 9; $(b)$ by construction of $\pi'_{\tau_t}(s')(a)$ in Eq. 6. Finally, assuming it is satisfied at $t = n - 1$, the induction step is proved for $t = n$, which ends the proof.

$\square$

## B  ALGORITHM

We briefly describe the EfficientZero algorithm from (Ye et al., 2021) and highlight the places where novelties are introduced in DEZ. In EfficientZero, there are several actors running in parallel:

- The Self-play actor fetches the current networks of the model (representation, dynamics, value, policy prediction and reward networks: $\mathcal{H}, \mathcal{G}, \mathcal{V}, \mathcal{P}, \mathcal{R}$). It samples episodes according to these networks, following the algorithm 1. Although the algorithm relies on episode sampling, the differences from the original version of EfficientZero, Ye et al. (2021) reside also in the episode post processing. After generating the trajectory, each episode is edited, associating each state $s_t$ with the executed action at time $t$ rather than the decided action at that time. This modification ensures that the replay buffer contains an episode that is effectively free of delays, allowing the utilization of existing learning procedures for effective learning. In addition, for the learner's sake, the self-play actors store statistics and outputs of the consecutive MCTS searches to the replay buffer.
- The CPU rollout workers (Ye et al., 2021) are responsible for preparing the batch contexts (selecting indexes of trajectories, and defining boundaries of valid episodes).
- GPU batch workers (Ye et al., 2021) effectively place batches on the GPUs and trigger the learner with an available batch signal.
- The Learner itself updates weights of the different networks using the several costs functions: the innovative similarity cost (for efficient dynamics learning), the reward cost, the policy cost, and the value cost.

As previously highlighted, it is important to note that, aside from modifications to the Self-Play actor and the testing procedure, our approach does not necessitate changes to the architecture. It is adaptable and can be integrated into any model-based algorithm.

Technically, the challenges were to incorporate the parallel forward of the four environments for effectiveness; to plan for the initial action queue based on the initial observation as described in Section 6; and to manipulate episodes according to the delay values in a complex structure of observations, actions, reward, and statistics of MCTS searches.

---

**Algorithm 1** DEZ: acting in environments with stochastic delays. PQR stands for Pending Queue Resolver (see Fig. 1)

---

$n \leftarrow 0$
**while** $n <$ STEPS **do**
    $\mathcal{H}, \mathcal{G}, \mathcal{V}, \mathcal{P}, \mathcal{R} \leftarrow \mathcal{H}_\theta, \mathcal{G}_\theta, \mathcal{V}_\theta, \mathcal{P}_\theta, \mathcal{R}_\theta$
    Sample new episode $(s_0, z_0, a_0, \ldots s_T, z_T, a_T)$
    Initialize queues:
    Default action queue: $[a_{-M}, \ldots, a_{-1}] = \bar{a}$. Delay queue: $[z_{-M}, \ldots, z_{-1}] = [M, \ldots, M]$.
    $t \leftarrow 0$
    **while** episode not terminated **do**
        Observe $s_t, z_t$
        Query from the delayed environment the estimated pending queue:
        $[\hat{a}_t, \ldots, \hat{a}_{t+z_t-1}] = \text{PQR}(a_{t-M}, \ldots, a_{t-1}, z_{t-M}, \ldots, z_{t-1})$
        $\hat{s}_{t+1} = g(s_t, \hat{a}_t)$
        $\vdots$
        $\hat{s}_{t+z_t} = g(\hat{s}_{t+z_t-1}, \hat{a}_{t+z_t-1})$
        $\pi_t = \text{MCTS}(\hat{s}_{t+z_t})$
        $a_t \sim \pi_t$
        Shift the action and delay queues and insert $a_t$ and $z_t$.
        $t \leftarrow t + 1$
    **end while**
    Post process the episode $(s_0, z_0, a_0, \ldots s_T, z_T, a_T)$ and compute effective decision times $\tau_0, \ldots, \tau_T$
    Add $(s_0, \tau_0, a_0, \ldots s_T, \tau_T, a_T)$ to the Replay Buffer.
    $n \leftarrow n + T$
**end while**

---

## C  EXPERIMENTS

### C.1  CONVERGENCE PLOTS FOR CONSTANT DELAY

Figure 5 gives the learning curves of DEZ together with the baselines for constant delay.

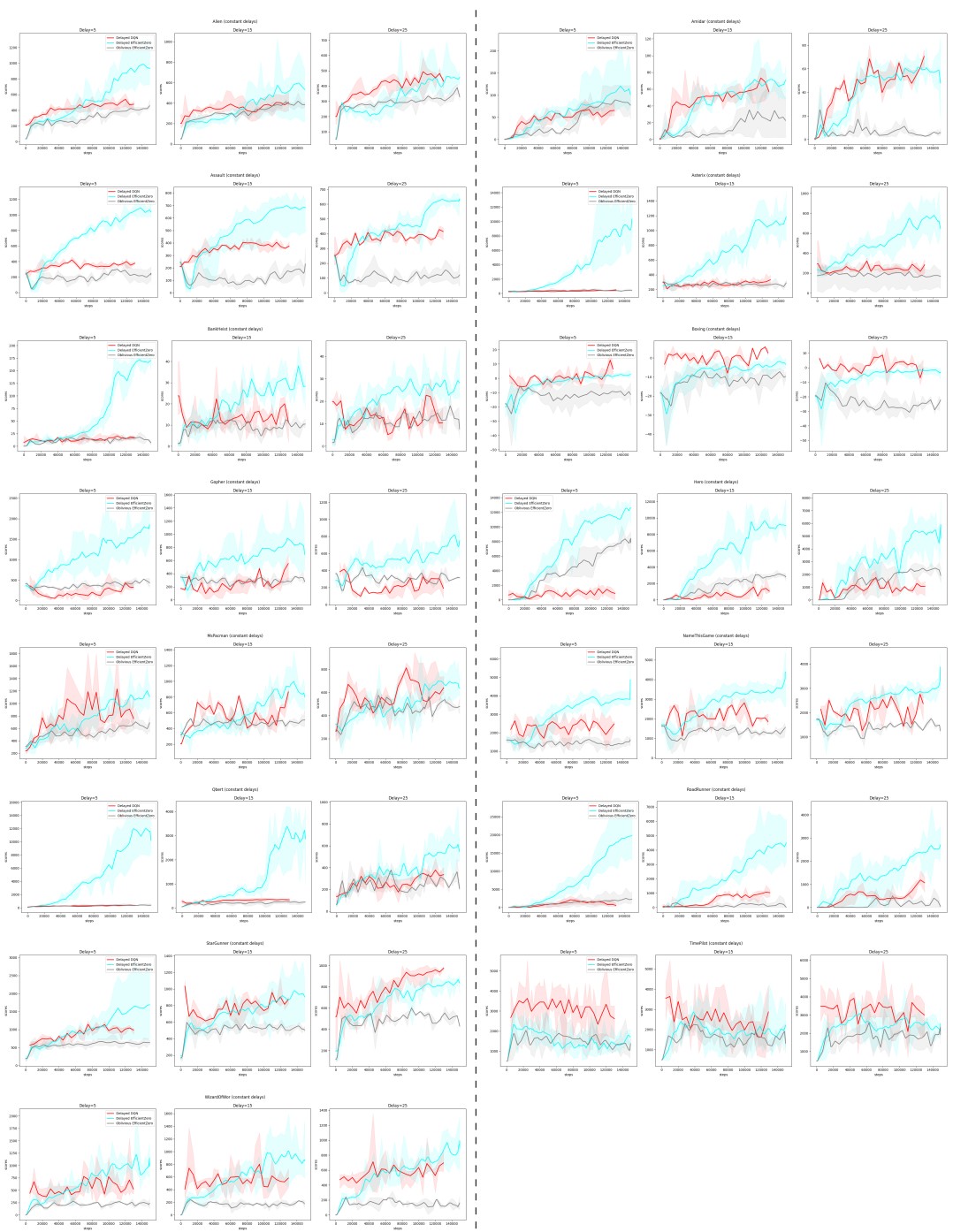

Figure 5: Convergence plots for 15 Atari games on constant delays in $\{5, 15, 25\}$.

## C.2 CONVERGENCE PLOTS FOR STOCHASTIC DELAY

Figure 6 gives the learning curves of DEZ together with the baselines for stochastic delay.

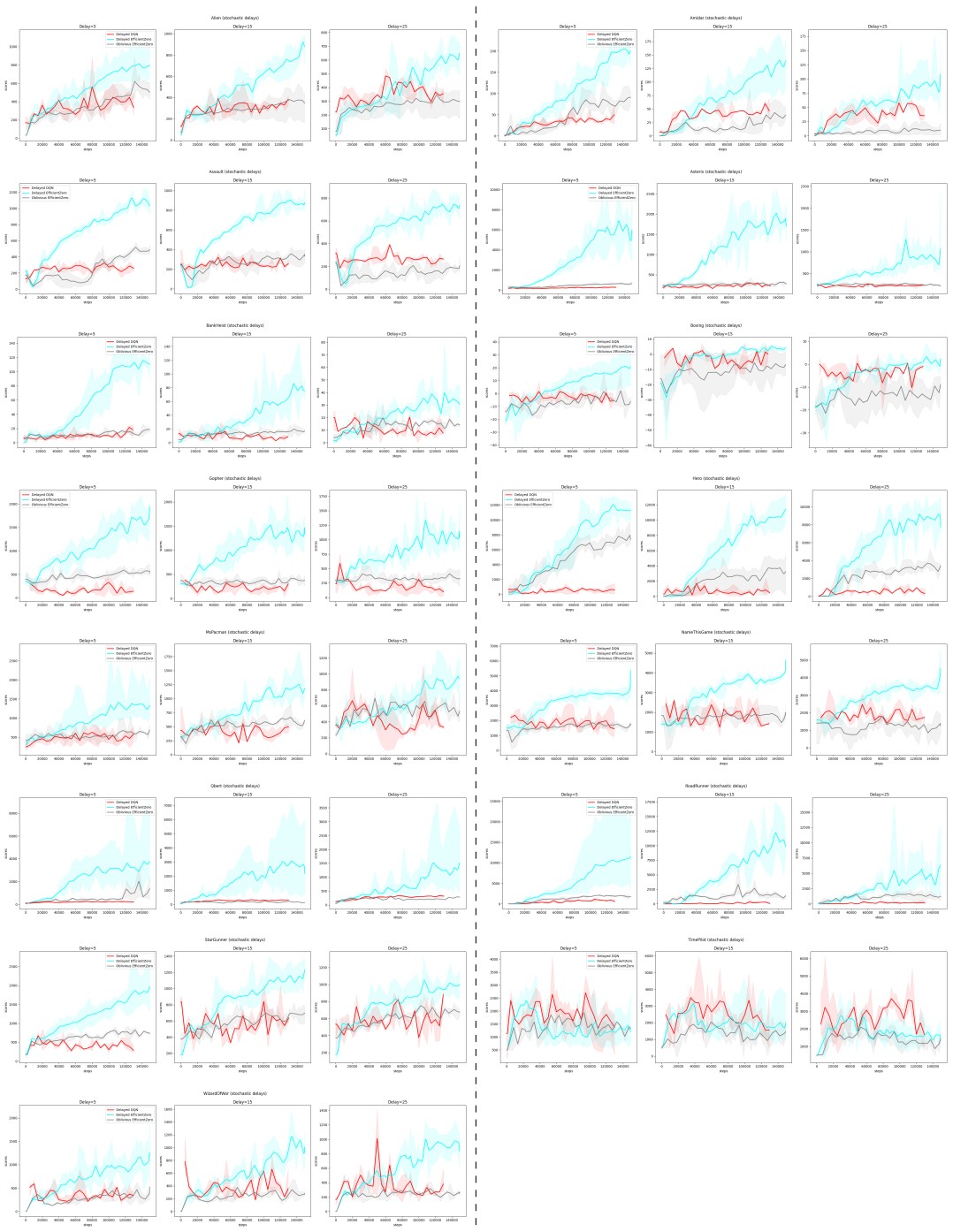

Figure 6: Convergence plots for 15 Atari games on stochastic delays with maximal delay $M \in \{5, 15, 25\}$ and probability $p = 0.2$.

## C.3 DEZ ON LARGE DELAYS

Here, we add results for very large delays in Asterix and Hero Atari games. As expected, the scores decrease as the delay increases, due to the complexity and error in planning.

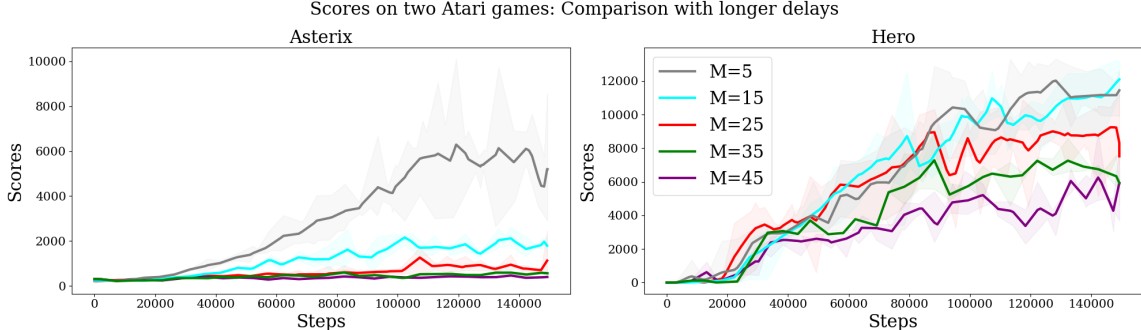

Figure 7: Scores on two Atari games with stochastic delays up to $M = 45$. Left: Asterix. Right: Hero.

### C.4 SUMMARY OF SCORES ON ATARI GAMES

We summarize the scores and standard deviations obtained for DEZ (ours) and baselines Oblivious EfficientZero of (Ye et al., 2021) and Delayed DQN of (Derman et al., 2021) on 15 Atari games. The following table shows scores on constant delays.

| Game | $m = 5$ | | | $m = 15$ | | | $m = 25$ | | |
|------|---------|---|---|----------|---|---|----------|---|---|
| | Obl. EZ | Delayed DQN | Delayed EZ | Obl. EZ | Delayed DQN | Delayed EZ | Obl. EZ | Delayed DQN | Delayed EZ |
| Alien | 335.5 ±24.88 | 484.33 ±73.67 | **993.25** ±311.5 | 287.86 ±30.57 | 421.33 ±58.0 | **512.0** ±109.75 | 278.8 ±29.0 | 454.0 ±120.5 | **497.25** ±185.75 |
| Amidar | 50.8 ±5.2 | 64.0 ±20.0 | **111.8** ±16.4 | 12.43 ±2.86 | 61.67 ±22.0 | **72.0** ±18.2 | 3.2 ±1.4 | 59.5 ±15.5 | **68.0** ±15.2 |
| Assault | 234.5 ±21.25 | 369.0 ±91.0 | **1056.5** ±281.25 | 208.6 ±33.2 | 371.0 ±93.67 | **674.75** ±59.75 | 179.75 ±20.0 | 389.5 ±107.5 | **636.6** ±92.4 |
| Asterix | 348.43 ±56.43 | 486.25 ±242.75 | **10758.2** ±5809.0 | 244.4 ±25.4 | 304.75 ±160.0 | **1160.71** ±447.86 | 186.4 ±15.4 | 248.33 ±136.33 | **673.4** ±314.8 |
| BankHeist | 13.25 ±4.25 | 17.0 ±13.5 | **180.2** ±33.2 | 11.4 ±2.2 | 13.67 ±12.67 | **35.6** ±17.4 | 12.5 ±3.0 | 14.17 ±10.83 | **29.4** ±14.6 |
| Boxing | -4.5 ±1.25 | **2.33** ±9.33 | 2.0 ±4.75 | -5.2 ±1.8 | **1.5** ±5.5 | -3.25 ±4.25 | -11.75 ±2.25 | **2.0** ±8.0 | -1.25 ±5.0 |
| Gopher | 373.5 ±43.0 | 294.67 ±190.33 | **1882.0** ±978.75 | 306.6 ±44.4 | 287.0 ±196.5 | **932.0** ±365.0 | 307.25 ±36.0 | 239.0 ±157.0 | **812.2** ±372.4 |
| Hero | 4867.6 ±497.4 | 948.67 ±1157.33 | **11233.2** ±1478.2 | 1949.4 ±156.0 | 717.0 ±1107.5 | **9630.4** ±1874.4 | 1364.5 ±273.5 | 843.5 ±1124.5 | **5908.8** ±1707.4 |
| MsPacman | 520.0 ±43.83 | 849.6 ±293.6 | **1091.4** ±338.8 | 386.0 ±40.2 | 665.2 ±178.2 | **934.5** ±360.5 | 367.5 ±35.0 | 616.67 ±91.67 | **716.2** ±377.6 |
| NameThisGame | 1707.17 ±151.67 | 2171.0 ±658.75 | **5021.4** ±1007.2 | 1725.4 ±149.0 | 2099.67 ±577.0 | **3926.17** ±697.17 | 1863.0 ±202.0 | 2083.0 ±568.25 | **3591.2** ±1033.4 |
| Qbert | 323.29 ±51.29 | 358.5 ±141.5 | **13085.5** ±2510.5 | 214.2 ±27.4 | 343.0 ±52.67 | **3241.6** ±1009.6 | 227.0 ±50.6 | 320.33 ±83.67 | **622.0** ±285.6 |
| RoadRunner | 1540.29 ±293.57 | 816.0 ±912.33 | **18485.25** ±4068.75 | 89.8 ±62.0 | 970.25 ±535.5 | **4636.0** ±948.4 | 106.0 ±71.5 | 1066.5 ±521.0 | **2642.2** ±846.2 |
| StarGunner | 631.0 ±64.14 | 992.0 ±232.33 | **1841.2** ±462.6 | 596.71 ±63.43 | 883.5 ±211.5 | **937.83** ±137.17 | 557.17 ±61.5 | **944.25** ±135.0 | 874.4 ±175.2 |
| TimePilot | 1876.71 ±340.29 | **2809.75** ±1265.5 | 2048.5 ±1564.75 | 2264.4 ±432.6 | 1981.25 ±908.75 | **2584.0** ±1277.25 | 2659.0 ±323.5 | **2871.67** ±1090.0 | 2615.4 ±1595.0 |
| WizardOfWor | 434.25 ±71.5 | 545.25 ±312.5 | **1150.8** ±664.6 | 348.8 ±60.8 | 525.67 ±226.67 | **1066.0** ±569.8 | 368.25 ±60.75 | 608.33 ±304.67 | **845.4** ±472.4 |

Table 1: Summary of mean scores on 15 Atari games with constant execution delay $M \in \{5, 15, 25\}$ on 32 episodes for each of the four trained seeds.

The following table shows scores on stochastic delays.

| Game | m = 5 | | | m = 15 | | | m = 25 | | |
| | Obl. EZ | Delayed DQN | Delayed EZ | Obl. EZ | Delayed DQN | Delayed EZ | Obl. EZ | Delayed DQN | Delayed EZ |
|---|---|---|---|---|---|---|---|---|---|
| Alien | 348.43 ±52.14 | 366.0 ±231.67 | **796.6** ±205.4 | 318.0 ±29.71 | 333.0 ±118.5 | **820.33** ±325.67 | 257.71 ±20.86 | 324.5 ±100.5 | **607.8** ±288.8 |
| Amidar | 41.5 ±6.17 | 37.0 ±21.67 | **220.25** ±43.0 | 23.4 ±5.8 | 48.5 ±27.0 | **145.67** ±30.67 | 7.33 ±2.83 | 40.0 ±17.5 | **95.8** ±32.4 |
| Assault | 376.0 ±20.14 | 267.0 ±140.67 | **1068.25** ±299.0 | 293.0 ±23.0 | 242.5 ±110.5 | **1132.0** ±286.33 | 203.4 ±30.0 | 251.5 ±98.5 | **790.6** ±191.8 |
| Asterix | 507.3 ±35.9 | 268.33 ±148.0 | **6535.5** ±4392.75 | 256.67 ±33.5 | 239.5 ±133.0 | **1225.0** ±934.33 | 233.17 ±16.33 | 234.5 ±168.5 | **636.0** ±396.6 |
| BankHeist | 15.0 ±3.33 | 13.0 ±12.67 | **140.0** ±30.6 | 16.17 ±2.33 | 6.5 ±7.5 | **91.67** ±26.67 | 14.0 ±2.5 | 9.0 ±12.0 | **27.33** ±16.0 |
| Boxing | -2.6 ±4.0 | -3.0 ±6.0 | **20.2** ±9.0 | -4.43 ±1.71 | -4.0 ±9.0 | **2.5** ±6.0 | -5.5 ±1.75 | -3.0 ±6.5 | **3.33** ±6.33 |
| Gopher | 424.38 ±35.62 | 145.0 ±127.33 | **1855.4** ±828.4 | 364.33 ±67.5 | 198.0 ±168.0 | **1633.0** ±709.0 | 347.4 ±31.8 | 130.0 ±118.5 | **1378.25** ±676.25 |
| Hero | 4813.88 ±475.75 | 492.0 ±856.0 | **11798.8** ±1539.6 | 2466.67 ±345.0 | 669.5 ±1006.0 | **11890.33** ±1372.0 | 1887.83 ±303.17 | 619.0 ±1027.5 | **8127.25** ±2681.25 |
| MsPacman | 530.0 ±64.62 | 444.0 ±276.0 | **1443.6** ±498.4 | 452.4 ±55.0 | 439.0 ±192.0 | **1173.0** ±612.33 | 397.17 ±66.17 | 399.0 ±162.5 | **940.6** ±455.8 |
| NameThisGame | 1933.33 ±127.33 | 1673.0 ±912.33 | **5395.0** ±1244.4 | 1991.0 ±160.0 | 1760.0 ±780.5 | **5029.33** ±1085.33 | 1513.44 ±243.0 | 1761.0 ±769.0 | **5027.4** ±1099.8 |
| Qbert | 548.8 ±207.0 | 208.67 ±113.33 | **3747.6** ±2603.8 | 154.0 ±26.0 | 296.5 ±88.5 | **4121.67** ±1829.0 | 176.75 ±30.5 | 303.0 ±78.5 | **1553.0** ±1275.5 |
| RoadRunner | 1254.0 ±161.33 | 668.0 ±649.5 | **9978.6** ±2914.6 | 876.4 ±166.8 | 227.0 ±515.5 | **12906.33** ±3311.0 | 488.75 ±113.62 | 260.5 ±352.0 | **2173.0** ±1103.0 |
| StarGunner | 700.2 ±71.6 | 378.5 ±234.0 | **1931.8** ±607.4 | 656.33 ±59.5 | 604.0 ±321.5 | **1235.33** ±249.67 | 666.14 ±59.86 | 621.5 ±356.0 | **939.33** ±199.33 |
| TimePilot | **2304.75** ±309.5 | 1529.67 ±972.67 | 1425.6 ±1347.8 | 1997.17 ±326.33 | 1905.25 ±1307.5 | **2128.0** ±1308.0 | 2405.5 ±379.83 | **2523.5** ±1488.0 | 1931.33 ±1778.0 |
| WizardOfWor | 541.8 ±125.8 | 288.5 ±247.0 | **1270.0** ±985.4 | 383.8 ±53.2 | 336.0 ±304.0 | **1175.0** ±900.0 | 409.5 ±59.67 | 294.5 ±233.5 | **1038.67** ±721.67 |

Table 2: Summary of mean scores on 15 Atari games with stochastic execution delay with maximal delay $M \in \{5, 15, 25\}$ on 32 episodes for each of the four trained seeds.

## D COMPUTATIONAL RESSOURCES

The computational costs associated with training DEZ in environments with delays increase in proportion to the delay values. This escalation arises from the multiple applications of the forward network during inference.

EfficientZero's architectural design harnesses the efficiency of C++/Cython for its Monte Carlo Tree Search (MCTS) implementation, intelligently distributes computation across CPU and GPU threads, thereby enabling parallel processing. Our experimental setup included two RTX 2080 TI GPUs. In the context of DEZ, each training run comprised 130,000 environment interactions and 150,000 training steps. We provide training duration statistics for the three delay configurations we employed:

For $M = 5$, the training duration exhibited fluctuations over a period of 20 hours. For $M = 15$, the training duration exhibited fluctuations over a period of 22 hours. For $M = 25$, the training duration exhibited fluctuations over a period of 25 hours.

The training duration of Oblivious EfficientZero is lightly shorter due to the omission of multi-step forward processing. For any delay value we tested, the training duration exhibited fluctuations over a period of 20 hours.

