# Tree Search-Based Policy Optimization under Stochastic Execution Delay

**David Valensi**
Technion
davidvalensi@campus.technion.ac.il

**Esther Derman**
Technion
estherderman@campus.technion.ac.il

**Shie Mannor**
Technion & Nvidia Research
shie@ee.technion.ac.il

**Gal Dalal**
Nvidia Research
gdalal@nvidia.com

## ABSTRACT

The standard formulation of Markov decision processes (MDPs) assumes that the agent's decisions are executed immediately. However, in numerous realistic applications such as robotics or healthcare, actions are performed with a delay whose value can even be stochastic. In this work, we introduce stochastic delayed execution MDPs, a new formalism addressing random delays without resorting to state augmentation. We show that given observed delay values, it is sufficient to perform a policy search in the class of Markov policies in order to reach optimal performance, thus extending the deterministic fixed delay case. Armed with this insight, we devise DEZ, a model-based algorithm that optimizes over the class of Markov policies. DEZ leverages Monte-Carlo tree search similar to its non-delayed variant EfficientZero to accurately infer future states from the action queue. Thus, it handles delayed execution while preserving the sample efficiency of EfficientZero. Through empirical analysis, we stress that none of the prior benchmarks consistently outperforms others across different delays. We demonstrate that our algorithm surpasses all benchmark methods in Atari games when dealing with constant or

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

 $\{5, 15, 25\}$ as in (Derman et al., 2021) **ED: is the full support in $\{5, 15, 25\}$ or are these max delay values?** . For both constant and stochastic delays, we refrain from random initialization of the initial action queue as in (Derman et al., 2021). Instead, our model determines the first $M$ actions. This is achieved through the iterative application of the forward and the policy networks. In practice, the agent observes the initial state $s_0$, infers the policy through $\bar{a}_0$, and predicts subsequent state $\hat{s}_1 = \mathcal{G}(s_0, \bar{a}_0)$. It similarly infers $\hat{s}_2, \ldots, \hat{s}_M$ and selects actions $\bar{a}_2, \ldots, \bar{a}_{M-1}$ through this iterative process.

EfficientZero sampled 100K transitions, aligning with the Atari 100K benchmark. Although our approach significantly benefits from EfficientZero's sample efficiency, the presence of delay adds complexity to the learning process requiring mildly more interactions – 130K in our case. Successfully tackling delays with such limited data is a non-trivial task.

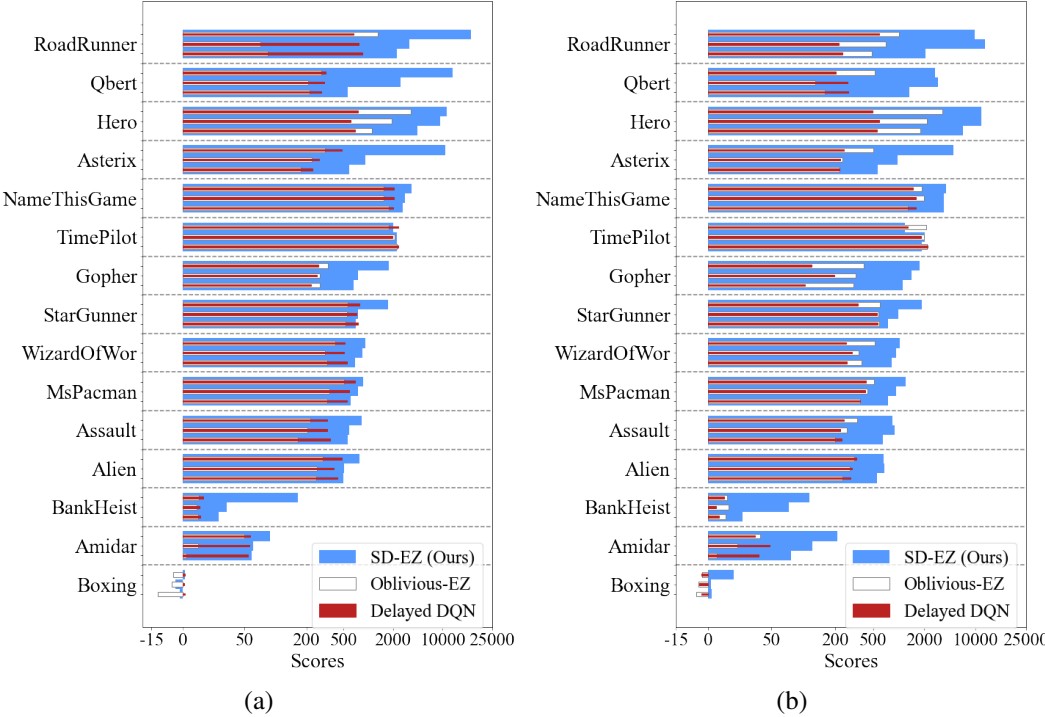

(a)          (b)

Figure 3: Average score on 15 Atari games and delays $M \in \{5, 15, 25\}$ over 32 test episodes per trained seed. Delays appear from low to high values for each game. (a) Constant delay value; (b) Stochastic delay value within $\{0, \cdots, M\}$.

To evaluate our theoretical findings, we subject DEZ to testing across 15 Atari games, each under both constant and stochastic delays. In either case, the delay value is revealed to the agent. We stress that the baseline scores are not consistent for the two types of delays we simulate. Indeed, on constant delays, Delayed-Q (Derman et al., 2021) tends to achieve higher scores than the oblivious version of EfficientZero, whereas Delayed-Q suffers more from stochastic delays than EfficientZero. As explained next, DEZ consistently achieves higher scores on both types of delays. A full summary of scores and standard deviations for our algorithm and baseline methods is presented in Appx. C.4, Tabs. 1 and 2.

### 6.1 CONSTANT DELAYS

We first analyze the scores produced by DEZ on constant delays and refer to Appx. C.1 for the convergence plots of all tested games.

Figure 3(a) shows that our method achieves the best average score in 39 of the 45 experiments. The oblivious version of EfficientZero confirms previously established theoretical findings (Derman et al., 2021, Prop. 5.2.): stationary policies alone are insufficient to achieve the optimal value. Given that Oblivious EfficientZero lacks adaptability and exclusively generates stationary policies, it is not surprising that it struggles to learn effectively in most games.

Our second baseline reference is Delayed-Q (Derman et al., 2021), acknowledged as a state-of-the-art algorithm to address constant delays in Atari games. However, it is worth noting