# OpenReview forum: "Tree Search-Based Policy Optimization under Stochastic Execution Delay"
_ICLR.cc/2024/Conference — ICLR 2024 poster_

### Official Review · Reviewer_Rzwe · 2023-10-31

**Soundness:** 2 fair
**Presentation:** 2 fair
**Contribution:** 3 good
**Rating:** 6
**Confidence:** 4

**Summary:**

The paper introduces(?) an MDP variant with stochastic execution delay. For the problem considered the authors show that  the optimal policy lays in a class of Markov policies. A variant of an MCTS algorithm, EfficientZero, is proposed that deals with the stochastic delay in the execution. Empirical evaluation are performed on a set of modified Atari benchmarks with superior results against some baselines.

**Strengths:**

The delayed MDP variant seems a relevant problem.

The theoretical insight for the considered variant is valuable, and the proposed algorithm extension seems reasonable.

**Weaknesses:**

It is not clear that this is a new MDP variant, or it is a particular version of the one with random delays considered in the Bouteiller article (cited in the paper) - with no observation delays.

The algorithm is described only in terms of differences to EfficientZero, which makes it hard to understand.

**Questions:**

The relation between the MDP variant considered here and the one discussed in the Bouteiller article should be made clear. I would prefer also some motivating discussion on the choice of dealing with the action delays. Personally, I would find more natural if the action delays would not be limited to discrete values, and being executed after the delay (in many physical systems, the evolution of the system is not restricted to discrete steps anyway, even if the decision are). This would avoid the drop and duplication of the actions. Also, we might not necessarily observe the realization of execution delay, unless we can reverse engineer it from a known transition model (when it is deterministic).

I can see arguments for the considered variant as well. Since the `proper' way to incorporate stochastic action delays in the MDP is not well established, I would like to see some (more) motivating discussion on the choice, and not only how it is handled.

While MuZero and (somewhat to lesser extent) EfficientZero are well known, it is still difficult to understand the new algorithm described only in terms of differences to EfficientZero. The algorithm should be described completely, while pointing out the relevant differences.

The experiments are fairly extensive, but there is the issue of lack of clear baselines when a new problem is proposed. Maybe adapting the Delay-Correcting Actor-Critic of Bouteiller (even if does not have an MCTS module)?

Minor: the initial state distribution (\mu) in the MDP tuple is not present in the SED-MDP definition. Is this an intentional omission?

Overall, I feel that there is a valuable contribution in the paper, but the presentation issues are fairly significant, and make the evaluation of the contribution difficult.

---

> ### Author Response · Authors · 2023-11-22
>
> Dear reviewer, thank you for your response. Please find our answers in the following.
>
>
> **W1: What is the difference from the formulation of (Bouteiller et al., 2020)?**
> In fact, there is a crucial difference between our work and (Bouteiller et al., 2020). The formulation in the latter included an augmentation of the state space with the action queue and delay values: X=S×A^K×N^2. This makes the decision process Markovian and therefore all standard results on non-delayed RL hold naturally at the expense of an exponentially larger new (embedded) state space. In our SED-MDP, however, the state space remains unembedded. We therefore devise the theoretical machinery needed to be able to prove that non-Markov policies are as beneficial as history-dependent policies, as long as they can be non-stationary.
>
> **Q1: What about continuous time delay and estimation of it without observing it?** We appreciate the useful real-world direction. We agree and added this direction to the summary as relevant future work.   We mentioned another approach when dealing with unobserved stochastic delays. Using a robust criterion with respect to multiple possible realizations of delays may lead to conservative policies, adapted to the delay complexity.
>
> **Q2: Can you add motivating examples to having stochastic delay?** Sure. Perhaps the most prominent example of why delay can be stochastic is in real world systems that rely on transmission of data. Often, there is interference in transmission that can stem both from internal or external sources. Internal sources may be due to sensing HW that can be dependent on the temperature and pressure conditions. Whereas external sources can stem in the case of a robot or AV whose policy infers actions remotely (e.g., in the cloud). We added this motivation to the introduction.
>
> **Q4: Can you baselines such as DCAC?**
> The Delay-Correcting Actor-Critic of Bouteiller et al., (2020) uses augmented state space, embedding the entire action queue, which we avoid by planning. Moreover, the DCAC algorithm does not support discrete action space and adapting Soft Actor Critic to these settings is a complex task as shown in (Zhou et al., 2022, Revising  discrete soft actor-critic) that requires significant modifications of the model structure, deviating from DCAC. With such a support to discrete action space, one could have used, for comparison, an adapted version of Soft Actor Critic for delayed environments - a Delayed Actor Critic with planning - but we believe this extension doesn't naturally align with the context.

---

> > ### Comment · Reviewer_Rzwe · 2023-11-22
> >
> > I appreciate the effort of the authors to improve on the presentation. I am raising my score slightly, although I still feel that it is a borderline paper.

---

### Official Review · Reviewer_zW4Z · 2023-10-31

**Soundness:** 3 good
**Presentation:** 2 fair
**Contribution:** 3 good
**Rating:** 6
**Confidence:** 3

**Summary:**

In the real world, state information may not instantly be observed, actions may not instantly be applied, and reward feedback may not be immediate for various environmental reasons. The authors tackle this problem, called stochastic execution delay in MDPs, through the introduction of a new formalism, SED-MDPs. They prove that during policy optimization it is sufficient to restrict the search to the set of Markov policies and present a novel algorithm called Delayed EfficientZero, which shows improved performance over EfficientZero and Delayed-Q on Atari benchmarks.

**Strengths:**

The authors motivate the problem well. It's easy to understand why stochastic execution delay is an important problem in RL. The contributions are also strong. The extension of ED-MDPs into SED-MDPs seems useful, and Theorem 4.2 is a nice theoretical result that shows Delayed EfficientZero is a well-principled algorithm for them. The results seem promising, given that Figure 5 is correctly labeled and not Figure 3.

**Weaknesses:**

There are some clarity issues with the experiments, particularly in Figure 3. It seems mislabeled. Figure 5 in the appendix suggests blue should be SD-EZ, red delayed DQN, and white Oblivious-EZ. It also looks like delays appear from "low to high" rather than from "high to low" as suggested in the caption. It would also be nice for comparison's sake to standardize the figure so that the same game appears on a single row in both columns.

Since SED-MDP is a new formalism, it may benefit the paper to include experiments in simpler domains. A central point of the paper seems to be constant vs. stochastic delays, but the experiments don't seem to show much of a difference between algorithms in the two environments for a given Atari game.

**Questions:**

My low score is down to the problems with Figure 3, which is central to supporting the claim about improved performance over Atari. If the authors can confirm my suspicions and commit to fixing the figure in the rebuttal, or at least clarify the situation, I will strongly consider raising my score (after considering other reviews as well).

---

> ### Author Response · Authors · 2023-11-22
>
> Dear reviewer, thank you for your response. Please find our answers in the following.
>
> **W1: Is the legend in Figure 3 wrong?** Good catch, the legend in Figure 3 was indeed wrong. Also, we added a table in the appendix, summarizing all the scores, including standard deviations for the four seeds.
>
> **W2: Can you make a each game appear in a single row in both columns?** Thanks for the suggestion. We have now standardized the game to appear in the same rows in both columns for comparison’s sake.
>
> **W3: Do we expect a difference between constant and stochastic delays?** We do not expect a significant difference in the score behavior between constant and stochastic delays. Since the delay process we used was not biased toward a single value, the effect of stochasticity in delays can be to ease the state prediction when delays become smaller. For both settings, the scores should decrease when the delay increases.

---

### Official Review · Reviewer_eDQ2 · 2023-10-31

**Soundness:** 3 good
**Presentation:** 3 good
**Contribution:** 3 good
**Rating:** 6
**Confidence:** 3

**Summary:**

The paper focus on MDPs in which there is a stochastic delay in the execution of the action selected by an agent acting in the MDP. Stochastic exectuion delay can result in the agent executing actions in the wrong state as the environment changes in real-time. Prior work focuses in fixed-delay times. This work provides a framework for situations in which the delay time is stochastic.

**Strengths:**

**Orignality:"" The approach introduced in the paper is a novel algorithm for a problem framed in a more realistic way than before.

**Clarity:** The paper is relatively clear and easy to understand. Some minor tweaks could be useful (see later.)

**Significance:** The approach proposed would be of interest to others working MDPs with stochastic delays.

**Quality:** The algorithm designed seems reasonable. The theoretical analysis looks sound however I did not thoroughly go through the proofs. The experiments chosen made sense however there are weaknesses in the results (see later.)

**Weaknesses:**

- I may be misunderstanding the graphs, but it looks as if SD-EZ scores worse than the other algorithms in most of the games in the plots in Fig 3a and 3b. Also, there are no confidence intervals or significance testing of any kind.

- The algorithmic description is slightly difficult to follow. Perhaps breaking down the data structures (lists, etc.) used into a list would help ease the process.

**Questions:**

No questions.

---

> ### Author Response · Authors · 2023-11-22
>
> Dear reviewer, thank you for your response. Please find our answers in the following.
>
> **W1 Mistake in Figure 3**
> Good catch, the legend in Figure 3 was indeed wrong; we fixed it in the revised version. Also, we added a table in the appendix, summarizing all the scores, including standard deviations for the four seeds.
>
> **W2 Algorithmic description**
> We added a description of the actors in the original EfficientZero model of Ye et al., (2021) along with a pseudo-code of the episode sampling procedure to Appendix B, to clarify the order of execution of the different components during episode collection.

---

> > ### Comment · Reviewer_eDQ2 · 2023-11-22
> >
> > I appreciate the authors' response. I am changing my score accordingly.

---

### Official Review · Reviewer_yeMT · 2023-11-03

**Soundness:** 2 fair
**Presentation:** 3 good
**Contribution:** 2 fair
**Rating:** 5
**Confidence:** 3

**Summary:**

This paper studies the delayed MDP problem, where stochastic delays are considered during the execution of actions. The authors introduce the concept of Stochastic Delayed Execution MDPs (SED-MDPs) as a solution to address random delays without relying on state augmentation. In particular, they show that optimizing within the set of Markov policies effectively reaches optimal performance.  Empirical validation across Atari games are performed under both constant and stochastic delay settings.

**Strengths:**

Standard RL assumes immediate availability of information for decision-making, overlooking delays prevalent in various real-world applications.  Existing approaches resort to state augmentation,  which is inefficient in handling exponential computational complexity and dependence on delay values, hindering its scalability to random delays. To address this issue, the authors propose Delayed EfficientZero, a delayed variant of EfficientZero, which is a model-based algorithm that optimizes over the class of Markov policies. The proposed method is able to accurately infer future states from the action queue, and thus handles delayed execution while preserving the sample efficiency of EfficientZero.

The authors establish the insight of optimizing within the set of Markov policies offers a more efficient and scalable solution compared to history-dependent policies. They introduce a model-based algorithm, namely, Delayed EfficientZero, that builds upon EfficientZero. The proposed algorithm yields non-stationary Markov policies, maintaining efficiency and scalability without making assumptions about the delay distribution.

Delayed feedback analysis is a rising topic in RL. Existing methods mainly focus on delays in states / trajectory / rewards. This work instead concerns delays in actions during policy execution, and this setting can be useful in practice.

**Weaknesses:**

1. This paper adopts the ED-MDP formulation of (Derman et al., 2021) that sidesteps state augmentation, and extend it to the random delay
case. This extension appears to be a direct application of the previous formulation, authors are expected to explain the technical challenges compared to the constant delay formulation, and be clear about their technical contribution in terms of this formulation.

2. This paper mainly develops based on the ED-MDP formulation of (Derman et al., 2021), random delay formulation in (Bouteiller et al., 2020),  and EfficientZero (Ye et al., 2021). The technical novelty in terms of the algorithm appears to be limited. Authors are expected to highlight the technical challenges and novelties.

3. While authors provide a thorough experimental study on 15 Atari benchmarks, they only consider small delays by setting $M = \{5, 15, 25\}$. It is desirable to see how the performance can be when delays are large.

**Questions:**

Under stochastic execution-delay MDPs (SED-MDPs), if at a specific time step (especially at the beginning of the game), there is no action available due to the delay, how does the proposed method execute the policy?

---

> ### Author Response · Authors · 2023-11-22
>
> Dear reviewer, thank you for your response. Please find our answers in the following.
>
> **W1 How is the extension technically challenging compared to the existing ED-MDP formulation of Derman et al., (2021)?**
> While both our work and that of Derman et al., (2021) tackle RL under delayed execution, we present new theoretical and algorithmic contributions. We understand the reviewer is asking regarding the theoretical part, so we answer it here. Unlike the fixed delay case of Derman et al., (2021), in an SED-MDP, the decision time of an executed action at $t$ is *random*. As a result, for a given policy, the process distribution can be different depending on the realization of the delay process. This is not the case of constant delay, as the SED-MDP process distribution stays the same once the policy is given. Yet, in Thm. 4.2, we show that *even* when the delay values are generated from a random process, for any history-dependent policy, we can achieve the same process distribution under a Markov policy. The technicalities of the proof require introducing the notion of *effective decision time* at time $t$, i.e., the effective time at which an action executed at $t$ has been drawn. Thus, instead of having $a_t\sim \pi_{t-m}$ for a constant execution delay $m$, we now have $a_t\sim\pi_{\tau_t}$, where $\tau_t}$ is the effective decision time – the time at which the action performed at time $t$ was previously decided. In some sense, the resulting process becomes a stopped process in the random delay case.
>
> **W2 Technical challenges and novelties in Delayed EfficientZero**
> As opposed to the Delay-Correcting Actor-Critic of Bouteiller et al., (2020), we leverage a model-based approach, which is different from their solution: Bouteiller et al., (2020) work on an augmented state space, a continuous action space, and effectively small delay values compared to ours (observation and execution delays are smaller than 7 in any of their wifi delay sampler).  The technicalities in our work also differ from Delay-Correcting Actor Critic as we describe here. We address our approach by leveraging the popular model-based approach of Ye et al., (2021), but this requires a significant effort to extend to the delayed setting. When delays exist in tree search-based algorithms, one has to perform these searches successively while keeping parallelism efficiency.  We do so by implementing a parallelized MCTS search for predicted states, game trajectory manipulation, and matching stochastic delays after observation are obtained as described in Section 5 and in the added Appendix B.
>
> **W3 Longer delay values**
> Thank you for your suggestion. We added experiments per your request on two Atari domains with larger delay values of {35,45}. The learning curves for Asterix and Hero can be found in the following link:
> https://ibb.co/q7npffz
> We also added them to Appendix C.3 in the revised version and the result explanations at the end of section 6.
>
> **Q1 What action to take when none is available?**
> As standard in control theory or previous delayed RL works, delay values are upper bounded (by, say, $M$) even though they are random. Thus, we specify an initial queue of $M$ actions to execute if none is available from the agent’s policy. Otherwise, especially when $t>M$, an action drawn from the agent’s policy is always available, and we use the latest executable action. This implies that some actions are duplicated when the realized delay value increases.

---

### Author Response · Authors · 2023-11-14
**Correction of Figure 3**

Dear reviewers, we apologize for the error in Figure 3's legend, which some of you have correctly spotted. We have now uploaded a revised version of the paper that includes a corrected Figure 3. For ease of reference, we also aligned the games in the two columns.

---

### Author Response · Authors · 2023-11-21
**Added results, tables, and algorithm**

Dear Reviewers and AC,
In response to a comment from Reviewer yeMT, we conducted additional experiments on two Atari domains with larger delay values of {35, 45}, as presented at the end of Section 6 and in the Appendix C.3, Table 7. The results demonstrate that the performance in such cases is primarily influenced by the specific characteristics of the environment.

We added the description of the Delayed EfficientZero algorithm in Appendix B. Here, we outline the different actors running concurrently in EfficientZero (Ye et al., 2021) and emphasize the modifications and challenges encountered when dealing with stochastic delays in a Tree-search based algorithm. Algorithm 1 in Appendix B provides details on the episode sampling procedure in Delayed EfficientZero. We appreciate the suggestions from Reviewer yeMT, which have significantly contributed to enhancing the clarity and completeness of the paper.

Additionally, as suggested by Reviewer eDQ2, we have added comprehensive tables of scores and standard deviations in Appendix C.4. Once again, we express our gratitude to Reviewer eDQ2 for this valuable suggestion.

The additions in the paper appear in blue, for your convenience.

---

> ### Comment · Reviewer_zW4Z · 2023-11-21
>
> I acknowledge that I've read the author comments and am raising my score accordingly.

---

### Author Response · Authors · 2023-11-23
**Slight modifications in the proofs of the two theorems**

Dear reviewers and AC,

To enhance clarity, we made slight modifications and corrections to the proofs of the two theorems.

In the proof of the first theorem, we adjusted the events $A_{2t+1}, a_{2t}, ...$ to present the delay and action realizations together. This alteration reflects the fact that action decisions depend on delay realizations. Although the proof underwent changes, it still follows similar principles.

For the second theorem, we positioned the definition of $\tilde{z}$ immediately after the equation.

In the proof of the second theorem, we introduced an induction base for $t=t_z$ as it is more natural to initiate the induction when the policy $\pi'$ effectively begins execution. We also made aesthetic modifications.

All modifications are highlighted in blue in the Appendix for your convenience.

---

### Meta-Review · Area_Chair_4irm · 2023-12-07

**Metareview:**

The paper considers stochastically delayed MDPs.  This is an important setting in practice that has not received a lot of attention by the research community.  It shows that searching in the space of Markovian policies is sufficient and therefore avoids an exponential blowup in comparison to state augmentation techniques.  There were questions about the novelty compared to Bouteiller's work, but the author response clarified the differences.  This work makes a clar contribution that advances the state of the art of algorithms for stochastically delayed MDPs.

**Justification For Why Not Higher Score:**

The reviews lack enthusiasm.

**Justification For Why Not Lower Score:**

This work makes a clear contribution that advances the state of the art of algorithms for stochastically delayed MDPs.

---

### Decision · Program_Chairs · 2024-01-16

Accept (poster)